# Enviro-HIRLAM model estimates of elevated black carbon pollution over Ukraine resulted from forest fires

Mykhailo Savenets[1], Larysa Pysarenko[1], Svitlana Krakovska[1], Alexander Mahura[2] and Tuukka Petäjä[2]

[1]Ukrainian Hydrometeorological Institute (UHMI), Kyiv, 03028, Ukraine
[2]Institute for Atmospheric and Earth System Research (INAR), Faculty of Science, Physics / University of Helsinki (UHEL), Helsinki, 00560, Finland

*Correspondence to*: Mykhailo Savenets (savenets@uhmi.org.ua), Alexander Mahura (alexander.mahura@helsinki.fi)

**Abstract.** Biomass burning is one of the biggest sources of atmospheric black carbon (BC), which negatively impacts human health and contributes to climate forcing. In this work, we explore the horizontal and vertical variability of BC concentrations over Ukraine during wildfires in August 2010. Using the Enviro-HIRLAM modelling framework, the BC atmospheric transport was modelled for coarse, accumulation, and Aitken mode aerosol particles emitted by the wildfire. Elevated pollution levels were observed within the boundary layer. The influence of the BC emissions from the wildfire was identified up to 550 hPa level for the coarse and accumulation modes and at distances of about 2000 km from the fire areas. BC was mainly transported in the lowest 3-km layer and mainly deposited at night and in the morning hours due to the formation of strong surface temperature inversions. As modelling is the only available source of BC data in Ukraine, our results were compared with ground-level measurements of dust, which showed an increase of concentration of up to 73% during wildfires in comparison to average values. The BC contribution was found to be 10-20% of total aerosol mass near the wildfires in the lowest 2-km layer. At a distance, BC contribution exceeded 10% only in urban areas. In the areas with a high BC content represented by both accumulation and coarse mode, downwelling surface long-wave radiation increased up to 20 W/m$^2$ and 2-m air temperature increased by 1-4°C during the midday hours. The findings of the case study can help to understand the behaviour of BC distribution and possible direct aerosol effects during anticyclonic conditions, which are often observed in mid-latitudes in the summer and lead to wildfire occurrences.

## 1 Introduction

Black carbon (BC) is the component of fine particulate matter (PM$_{2.5}$) considered as one of the contributors to climate forcing next to carbon dioxide (Bond et al., 2013, Kurganskiy et al., 2015) and has a highly probable harmful health impact (Janssen et al., 2011; WHO, 2012; O'Dell et al., 2020). BC is formed as a product after incomplete combustion of biomass and fossil fuels (e.g., Forbes et al., 2006, Bond et al., 2013). A large amount of BC is emitted into the atmosphere from biomass burning (Konovalov et al., 2018) as a part of total chemical species flux during wildfires (Amiro et al., 2001; Barnaba et al., 2011;

Virkkula et al. 2014a), which causes elevated pollution concentrations around burned areas (e.g., Virkkula et al. 2014b; Wu et al., 2018; Castagna et al., 2021). BC content and different aerosol constituents in case of huge emissions are frequently estimated by using atmospheric modelling (Hodzic et al., 2007; Bessagnet et al., 2008; Konovalov et al., 2018; Singh et al., 2018; Magalhaes et al., 2019; Kostrykin et al., 2021) and sometimes by in-situ measurements (Yttri et al., 2007; Eleftheriadis et al., 2009; Singh et al., 2018; Jia et al., 2021). In contrast to other aerosol compounds, BC typically causes a positive radiative

forcing (Bond et al., 2013; Stjern et al., 2017), whose intensity depends on the particle size (Matsui et al., 2018). Consequently, the heating effect is generally observed from wildfire emissions (Kostrykin et al., 2021), and from anthropogenic sources (Zhuang et al., 2019). Sometimes a cooling effect was also detected (Ma et al., 2018). All of these effects, however, are localized (Modak and Bala, 2019). In Ukraine, there are no observations of carbonaceous aerosols in the atmosphere. Moreover, only a few studies have mentioned Ukrainian territory with elevated BC content during wildfire events (Pavese et

al., 2002; Yang et al., 2017).

As a starting point for the current case study, the forest fire events occurred in summer 2010 at the centre of the European territory of Russia with strong wildfire emissions. These emissions were supported by extremely hot weather and rainless conditions. Several studies discussed different environmental aspects of this event, including aerosol distribution (Konovalov

et al., 2011; Witte et al., 2011; Galytska et al., 2016; Galytska et al., 2018), radiative effects (Chubarova et al., 2012), and air temperature changes (Pere et al., 2014). The emissions were detected and influenced atmospheric composition in Finland as well (Leino et al., 2014). The most severe period was related to the August fires with a lot of pollution transported towards Ukraine (Galytska et al., 2016). The episode ended after 18th August when the pollution levels returned to typical values for these geographical regions (Witte et al., 2011).


In more detail, Galytska et al. (2018) focused on the analysis of summer 2010 wildfires events and studied aerosol content changes for the period 1 July – 20 August 2010. This study used satellite data from the Moderate Resolution Imaging Spectrometer (MODIS) and the Cloud-Aerosol Lidar with Orthogonal Polarization (CALIOP) for detecting burned areas, aerosols vertical profiles, and clouds. Data from ground-based sun photometers of Aerosol Robotic NETwork (AERONET),

measuring column-integrated aerosol optical depth (AOD), were validated against the satellite measurements. The Hybrid Single-Particle Lagrangian Integrated Trajectory (HYSPLIT) model was used to simulate air mass backward trajectories (with vertical levels of 0.5, 1.5, 3, 4, and 5 km), and analysis of the meteorological situation and air masses transport were performed with the help of a series of synoptic weather maps. According to Galytska et al. (2018), the maximum AOD above Moscow (Russia) on 7th August was associated with airflows from the central part of Russia at levels of 0.5–1.5 km passing through

fire areas. In Kyiv, the maximum AOD was registered on 15th August (up to 4 km level) with air movements from the forest fire regions. The simulated air trajectories referred to anticyclonic movements in both cases. These conditions facilitated the stagnation of air and the accumulation of pollutants in the region of interest. In Sevastopol, the maximum AOD was observed on 16th August within the layer 0.5–5 km due to air masses transport from the territory of active fires.

The spatiotemporal distribution of trace gases and aerosols at the ground level was analysed by Konovalov et al. 2011. The total CO, $PM_{10}$, and $O_3$ concentrations were analysed using satellite measurements and the CHIMERE chemistry transport model (Konovalov et al., 2011) with the most attention paid to the Moscow region. It was found that daily mean CO and $PM_{10}$ concentrations reached values up to 10 and 700 μg/m$^3$ respectively. $O_3$ concentrations were episodically very large (up to 400 μg/m$^3$) even after emissions significantly decreased. It was estimated that approximately 10 Tg CO was emitted in the Moscow

region during the 2010 heat wave (more than 85% of the total annual anthropogenic CO emissions). Aerosol particles emitted from summer 2010 wildfires caused shortwave direct radiative effects (Pere et al., 2014). Significant reduction of diurnal average solar radiation was found to be at the ground up to 80–150W/m$^2$. The resulting feedbacks lead to a cooling of the air up to 1.6°C at the surface and up to 0.1°C at altitudes of 1.5–2.0 km.

Described wildfire events frequently occur in Ukraine or in the territories of neighbouring countries that develop into elevated regional pollution events. Unfortunately, the existing official Ukrainian air quality monitoring network (UA-AQMN) does not provide any measurements of BC. Hence, modelling remains the only possible way to estimate the spatio-temporal variability of BC in Ukraine and to explore the consequences of elevated pollution episodes and assess the short-term impacts in the region. The lack of observation capacity on BC on a national level caused a gap in knowledge on how BC is distributed; what

are the impacts on local ecosystems; and which mitigation measures are needed to improve the situation. This study is the first in Ukraine which describes the elevated spatio-temporal BC content on the three-dimensional scale, explores the distribution of BC in particle sizes, and compares the BC ratio between other aerosol compounds in the atmosphere with an emphasis on extremely hot weather conditions. Furthermore, we aimed to estimate the impact of wildfire emissions on the surface by considering direct aerosol effects as the response of radiative and temperature regimes varying in different regions and

depending on the ratio of aerosol compounds.

    We explored the BC concentrations in Ukraine during the intensive biomass burning in the region in August 2010. We deployed the Environment – High Resolution Limited Area Model (Enviro-HIRLAM) modelling system in the analysis. It is a fully online coupled (integrated) numerical weather prediction (NWP) and atmospheric chemical transport (ACT) modelling

system for research and forecasting of meteorological, chemical, and biological weather (Baklanov et al., 2017). These simulations will help us to understand the temporal and spatial BC distribution after wildfire emissions in the case of summer anticyclonic conditions leading to frequent wildfires occurrence in mid-latitudes and resulting in elevated pollution levels. Here we initialized a case study that was carried out for elevated pollution episodes in August 2010 over Ukrainian territory caused by atmospheric pollution transport from the severe forest fires occurred in the central part of Russia. BC concentration

and its distribution were estimated for an extended period 2-18 August 2010, considering the extreme pollution episode lasted 7-17 August 2010. Direct aerosol effects on downwelling surface short-wave/ long-wave radiation and 2-m air temperature are also discussed.

## 2 Data and methods

### 2.1 Enviro-HIRLAM setup

The Enviro-HIRLAM system consists of two main blocks. The HIRLAM model itself as an NWP model, which is used for research and operational purposes (HIRLAM-5 Scientific Documentation, 2002). The Enviro-components are integrated into the NWP model (Baklanov et al., 2017). These components include a variety of atmospheric chemistry schemes, which simulate tropospheric sulphur cycle chemistry, gas-phase chemistry, photolysis rates, heterogeneous chemistry, aerosol formation and dynamics, wet and dry deposition and include different feedback mechanisms (direct, semi-direct, indirect). For

modelling of fluxes of BC, the Enviro-HIRLAM was deployed for the Northern Europe and Arctic regions (Kurganskiy et al., 2015; Nuterman et al., 2015).

In this study, the model domain covered almost all the European territory and is enlarged for considering the atmospheric circulation in the middle latitudes. It consisted of 500x400 grid points with a 15-km horizontal resolution and a time step of

120 sec. The area of interest is within 20–45°E and 40–60°N (Fig. 1). The model output was saved at every 3 h interval.

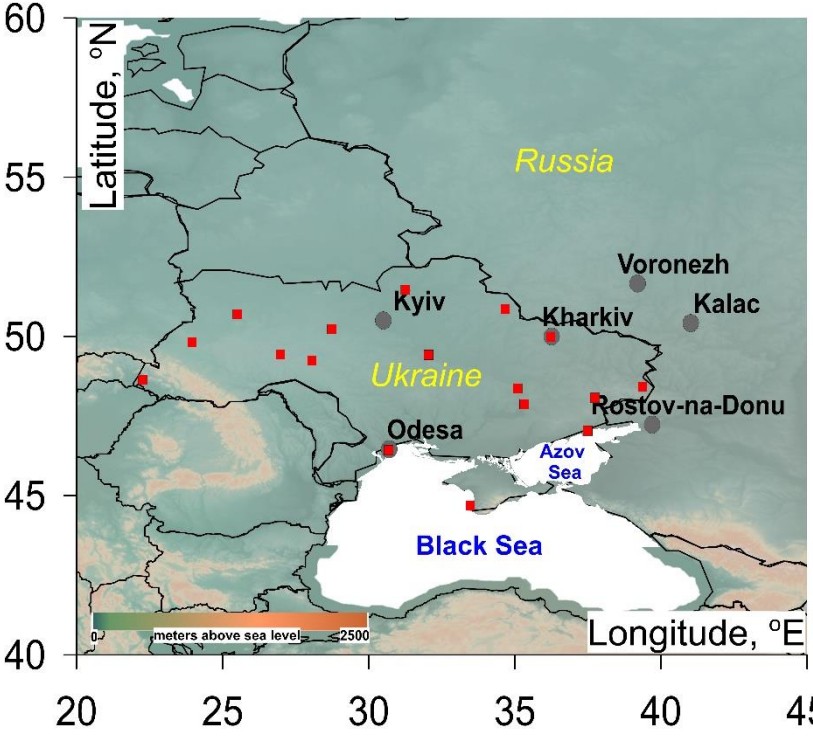

**Figure 1: The area of interest and geographical locations of the radiosounding stations (black circles - Kyiv, Odesa, Kharkiv, Voronezh, Rostov-na-Donu, Kalac) and air pollution monitoring stations (red squares) used in this study.**


The vertical structure included 40 model levels (up to 10 hPa) with a more detailed resolution in the boundary layer with 22 model levels from the surface up to 500 hPa. This provided a great opportunity to study the BC vertical atmospheric transport. The model was run as follows: first, the reference (or control) run (REF); followed by the run with direct aerosol effects (DAE) included.


The initial and boundary conditions (ICs/BCs) for meteorology included components of wind speed, air temperature, and specific humidity extracted at all model levels and at every 3-hour interval from the ERA5 model archives at the European Centre for Medium-Range Weather Forecasts (ECMWF). The sea surface temperature and the conventional Binary Universal Form for the Representation (BUFR) of meteorological data observations for data assimilation were extracted at every 12-hour and 3-hour interval, respectively. The ICs/BCs for gas and aerosol concentrations included 3D fields of mixing ratio of aerosol compounds (dust, hydrophilic and hydrophobic organic matter and black carbon, sulphates) as well as gases ($O_3$, $SO_2$, $NO_2$, NO, hydrogen peroxide ($H_2O_2$), hydroxyl radical (OH), nitrate radical ($NO_3$), hydroperoxyl radical ($HO_2$), dimethyl sulfide (DMS) were extracted at all model levels and at every 3-hour interval from Copernicus Atmosphere Monitoring Service (CAMS) of ECWMF.


A suite of emission inventories (EIs) was utilized in the model runs, including anthropogenic and biomass burning (wildfires). In more detail, the EIs given in geographical latitude/longitude domain were as follows: 1) for the global biomass burning (wildfires) emissions from IS4FIRES (Sofiev et al., 2012), 2) gridded emissions of gases and aerosols of $SO_2$, $NO_x$, $NH_3$, non-methane VOC, BC, OC, $PM_{2.5}$, $PM_{10}$, CO, $CH_4$ for specific Selected Nomenclature for Air Pollution (SNAP) codes such as industry, transportation, agriculture, etc.) from Evaluating the Climate and Air Quality Impacts of Short-Lived Pollutants (IIASA's ECLIPSE, 5th version), 3) shipping emissions, including $SO_2$, BC and OC and, emissions of DMS from Nightingale et al. (2000). The emission pre-processor included both vertical and temporal profiles for the model setup.

The spatial analysis of model output was carried out considering all grid cells without spatial averaging and interpolation. It enabled the detection of concentration changes within each grid caused by anticyclonic air movements. Evaluation of accumulated BC impact was performed by time integration of near-surface BC concentration at the lowest (near-surface) model level for the studied period. Therefore, the summed values represent the total amount (for the Aitken, accumulation, and coarse modes in ppb) transported by air movements through grid cells near the surface.

## 2.2 Additional data for the analysis

Unfortunately, national UA-AQMN does not provide measurements of BC, moreover PM is not measured either. Dust is the only available pollutant that could be compared to the modelled aerosol species. The dust is measured at more than 120 monitoring sites, and measurements contain all coarse aerosol particles regardless of their origin (Nadtochii et al., 2019). The dust concentration on these sites is measured using the method of weighing the total suspended particles (RD, 1991).

Therefore, it is complex to compare results with ground-level measurements accurately. UA-AQMN was established several decades ago and has been under continuous development and expansion. Therefore, the majority of the monitoring sites are situated in cities. There is a large number of anthropogenic sources such as factories, thermal power stations, roads, etc. in the city environment. Hence, we performed daily averaging to calculate an integral value for each city based on several monitoring stations. Such an approach will improve the signal-to-noise ratio in a time series. In our study, dust mass concentration data was selected from the year 2010 in 20 Ukrainian cities, which are approximately geographically equally distributed within the country. It was done for purposes of intercomparison between western-eastern-southern-northern territories.

The upper air sounding data from the Wyoming University database (weather.uwyo.edu/upperair/sounding.html) were used to detect temperature inversions. The air temperature vertical profiles were analyzed for 2–18 August 2010 at the following sounding stations: Kyiv (station code 33345), Kharkiv (34300), Odesa (33837), Rostov-na-Donu (34731), Kalac (34247) and Voronezh (34122). Unfortunately, there is a limited number of the sounding stations in the region of interest. Hence, Voronezh and Kalac were selected because these are situated within a radius of less than 400 km away from the forest fires area (being the nearest sounding stations to wildfires). Kharkiv and Rostov-na-Donu were selected for estimation of temperature inversion impact at some distance (within a radius of up to 600 km) away from the fires. Kyiv and Odesa were chosen as these are relatively distant (within a radius of up to 1000 km) to wildfires emission sources. The air temperature vertical profiles were considered for up to 3.5 km above the ground surface at 00 UTC.

## 3 Results and discussion

### 3.1 Synoptic weather situation and the dispersion of wildfire emissions in Ukraine during August 2010

According to the Climate Forecast System (CFS) Reanalysis (source: www.wetterzentrale.de) of the 500 hPa geopotential maps over Europe, a blocking anticyclone caused severe hot weather, lack of precipitation, which lead to the occurrence of wildfires. The period lasted from the end of June to the second half of August 2010 over Eastern Europe and the south-western regions of Russia (see example Fig. A1). Hot air masses from Central Asia penetrated the territories on the north-west, and the anticyclone was detected throughout the whole troposphere before the highest pollution levels were transported and distributed out of burning areas. Continuous extreme weather and clear sky conditions together with the highest insolation in the middle latitudes caused the dominance of high temperature and low humidity regimes. These led to the most favourable conditions for drought formation that played a crucial role in initiating the wild fires and contributed to their rapid spread.

At the beginning of August 2010, before the wildfires, the near-surface BC content over Ukraine was low and mostly it was represented by the accumulation mode with values of 1-8 ppb. The highest accumulation mode BC concentrations reached

value of 16 ppb and were observed at the near-surface level near small local wildfires in the central part of Ukraine. The BC
concentrations in the Aitken and coarse modes did not exceed value of 0.5 ppb.

The main source of wildfire emissions was located outside of the Ukraine's territory and consisted of several burning areas (see Fig. 2b). The observed anticyclonic conditions influenced the formation and development of spatio-temporal patterns for BC atmospheric transport and dispersion. The time-series for each grid point consisted of two maxima. These were related to the observed atmospheric circulation patterns. A typical clockwise air movement for anticyclones in the Northern Hemisphere caused an intensive atmospheric transport towards Ukraine during two periods: 7-8 and 13-16 August 2010. During these periods, the elevated concentrations were also observed in the northern regions of Ukraine (as shown in Fig. 2).

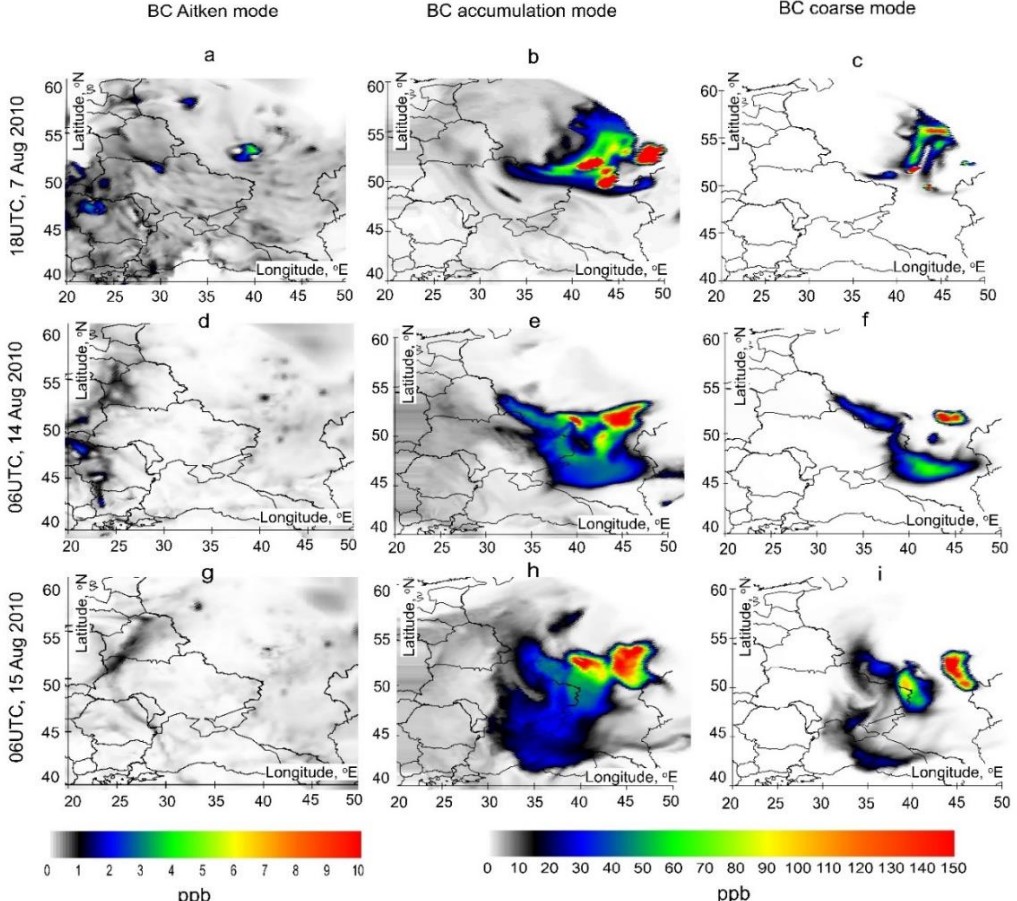

**Figure 2: The spatial distribution of the BC for the Aitken (a,d,g), accumulation (b,e,h) and coarse (c,f,i) modes in the days of air movement towards Ukraine with the highest BC content near the surface.**

For these days, the highest BC values for the accumulation mode near the surface exceeded values of 400-600 ppb near the burning areas and values of 70-150 ppb over the selected territories of the eastern, central, and northern parts of Ukraine. For

the coarse mode, the corresponding values were 300-450 ppb and 20-60 ppb for the same territories, respectively. The coarse mode BC mainly deposited close to the burning areas and only a small fraction was transported more than 500 km away from them (Fig. 2). In contrast, the accumulation mode BC was transported far from the burning areas by the prevailing wind, and it was the dominant BC mode at a distance of ca. 1000 km (Fig. 2). The highest values for Aitken mode BC did not exceed 6 ppb near the fires, and the simulated values were lower than 2 ppb at a larger distance from the burning areas. Moreover, the highest values of 10 ppb of the Aitken mode BC were intermittently observed over urban areas. Due to the geographical location of the wildfires, maxima in the BC concentration time-series shifted from the east and north-east to the west and south-west regions of Ukraine. Such a shifting depended on anticyclonic air mass movements. Nevertheless, during 7–8 and 13–16 August, the higher values in the western part of Ukraine were observed within 24 h after the maxima emerged in the east. During unfavourable circulation conditions, the pollution plumes were transported and dispersed at distances of more than 2000 km away from the original wildfire areas. The dominated hot and dry weather conditions interrupted in the second half of August 2010. It occurred when the blocking anticyclone weakened, and a cyclone arrived from the western sector.

In general, wildfire emissions had a large cumulative effect in the near-surface layer concentrations. The total accumulated amount of BC for the period 3–18 August 2010 reached 13500 ppb for the accumulation mode and 2200 ppb for the coarse mode, respectively, in the lower tropospheric layer near the burning areas (Fig. 3). The total accumulated amount of BC Aitken mode caused by wildfires was about 15-30 ppb, whereas the maximum accumulated impacts were observed in the cities (see e.g., Aitken mode in Fig. 3). A large amount of combustion products was transported through the atmosphere to the south-west and deposited over territories of the Eastern Ukraine, the Azov and Black Seas. The integral values of BC for these territories exceeded values of 800 and 150 ppb for the accumulation and for the coarse modes, respectively. This is seen in Fig. 3, where the regions were affected by intensive deposition processes. Due to the smaller sizes of the particles, the accumulation mode had a larger spatial extent and a smoother distribution than the coarse mode.

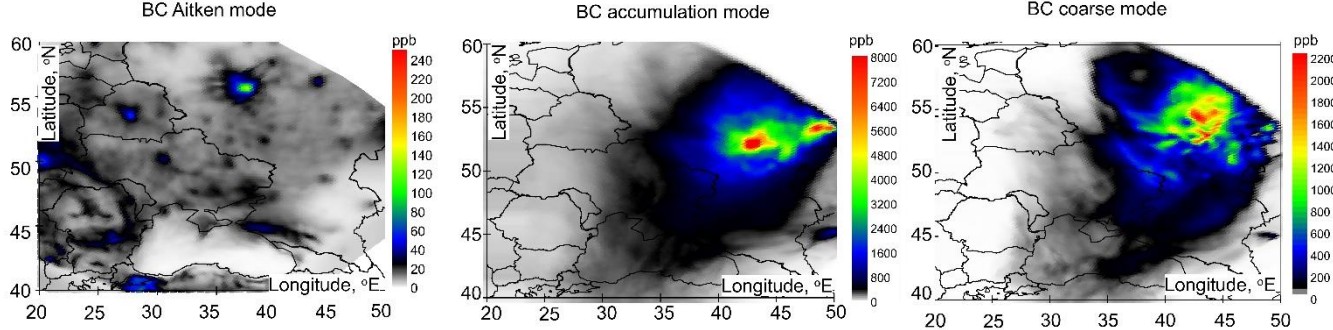

**Figure 3: The integral value of the near-surface BC concentrations for the Aitken, accumulation and coarse modes for the period 3-18 August 2010.**

For the studied period, in general, the ground-level dust measurements showed elevated concentrations presumably linked with the forest fires. Except for the northern part of Ukraine, almost all geographical regions experienced clear maxima of ground-level dust concentrations during August 2010. The highest concentrations were observed in the Eastern territories, where the values exceeded the urban background values in August. In this case, the integral value of the near-surface BC concentrations for the coarse mode was higher than 250 ppb. In the cities in the east Ukraine, the dust concentrations were 27–47% higher than the average dust content in 2010. Moreover, these values were 23–72% higher than the multi-year average concentrations for the month of August. Overall, the dust content in the east Ukraine was 17-60 ppb, which was higher than what usually occurs in August.

A large difference in the dust content between August and other months was observed on the seashore of the Azov Sea and in the central part of Ukraine. The concentrations were higher by 4-20 ppb than usually in the same month. A majority of cities in the central part of Ukraine showed 17–73% higher dust concentrations than average in 2010 and, concurrently, 8–45% higher than usually in August. In the western parts of Ukraine, the integral values were lower than 100 and 500 ppb for coarse and accumulation modes, respectively (Fig. 3).

**3.2 Diurnal variability and vertical distribution of BC in the region**

During 3-18 August at nighttime, the whole European territory of Russia, and the central and eastern territories of Ukraine were characterized by a presence of surface air temperature inversions. This was indicated by diurnal variation of BC in the lowest 500 m layer. The deepest and strongest inversions, up to 655 m depth and up to 12.5°C, were observed during 4–7 August and on 16 August (see Fig. A2). At 500 m level, the air temperature was warmer by 10–12°C than at 2 m height. On the other nights, the inversions were weaker with an average difference of only 3–4°C in the 2-500 m layer.

In our study, the diurnal variations of the BC content in the boundary layer were well identified. Maxima was often observed at nighttime and during morning hours, whereas the daytime pollution levels were low in the lower troposphere (see Fig. 4). These diurnal variations occurred due to radiative surface cooling during summer nights, and especially, in the case of blocking anticyclones. Such anticyclones facilitated the formation of air temperature surface inversions. Together with the intensification of downward air movement at night, BC from the whole lower and middle tropospheric levels is accumulated within the boundary layer, reaching elevated values there. During the daytime, the processes of less intensive air descending and turbulence increasing resulted in a more homogeneous vertical distribution. These processes led to a decrease in near-surface BC content. As it happened throughout the Ukraine's territory with anticyclonic weather, spatial distribution of BC at the separate level in the lower troposphere was dominated by horizontal dispersion. In the middle troposphere, any diurnal variations of BC content are negligible because surface inversions do not reach these altitudes.

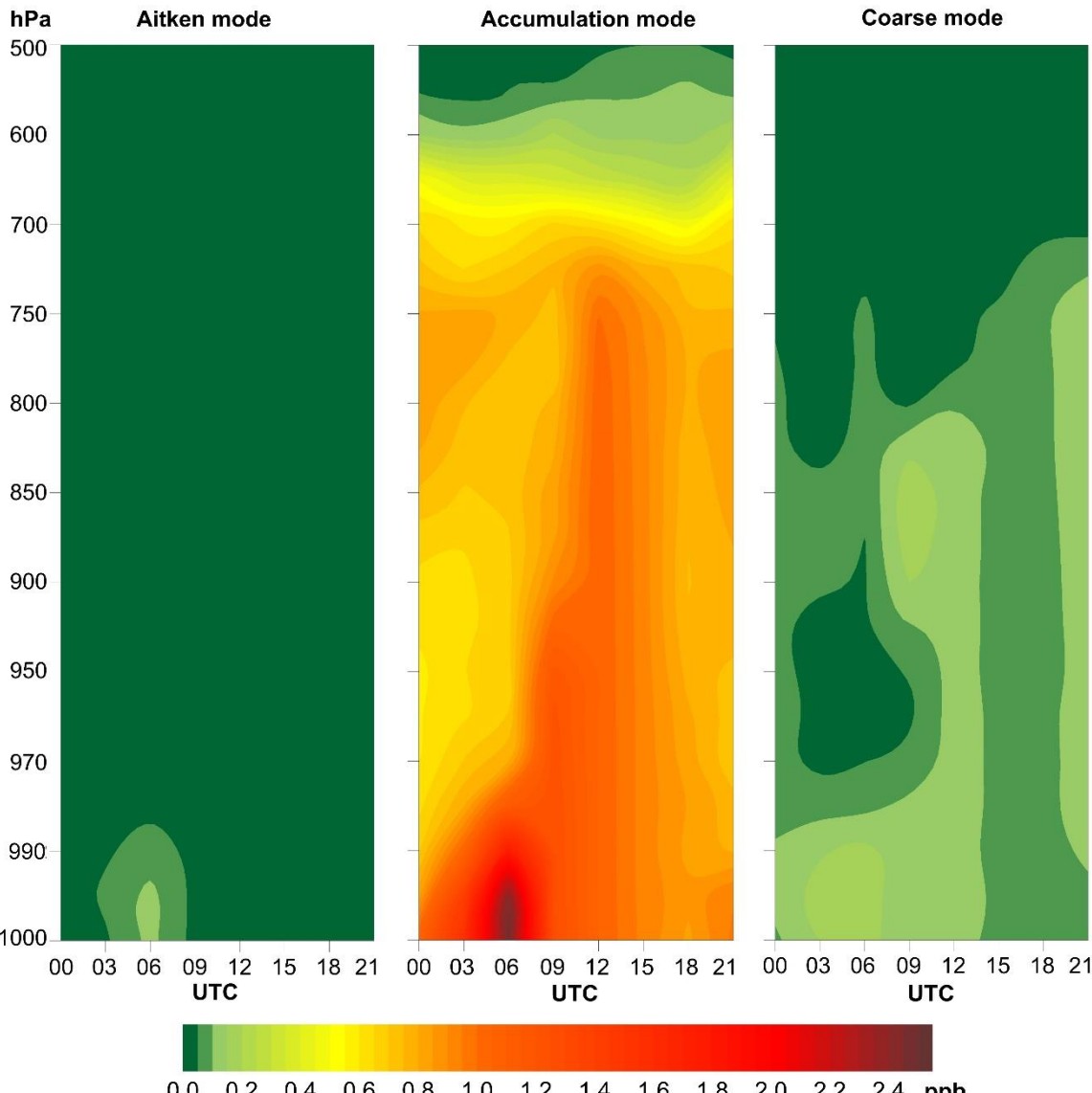

**Figure 4: Diurnal variability of the BC concentration at the pressure model levels for the Aitken, accumulation and coarse modes on example of the Kharkiv sounding station. The values were calculated as a mean diurnal cycle for the period of study.**

The vertical distribution of the coarse mode BC and accumulation modes was well detected in the lowest 3-km layer with the maximum observed in the boundary layer. Approximately at the 700 hPa level, BC concentration for both accumulation and coarse modes started to decrease very rapidly (Fig. 5). The levels of 660-630 hPa for the coarse mode and 590-450 hPa for the accumulation mode were identified as the highest altitudes where the influence of wildfire emissions was detected constantly

during the daytime. The elevated BC concentrations were rarely detected at 590 hPa (for the coarse mode) and at 400 hPa (for the accumulation mode) levels, where the concentrations remained below 0.1 ppb. In other words, in the middle troposphere, diurnal variations of BC content were negligible because surface inversions did not reach these altitudes.

265

In contrast to the coarse and accumulation modes, the Aitken mode was observed throughout the entire troposphere up to 200 hPa level (Fig. 5), however, the wildfire emissions of the Aitken mode did not dominate over the anthropogenic emissions. The Aitken mode rarely exceeded 0.2 ppb at the 950 hPa level. Therefore, concentrations of more than 1 ppb were observed only near the surface and had a clear maximum over urban areas (see Fig. 3). This is the reason why the Aitken mode near the
270    surface was higher at more distant Odesa than Kalac.

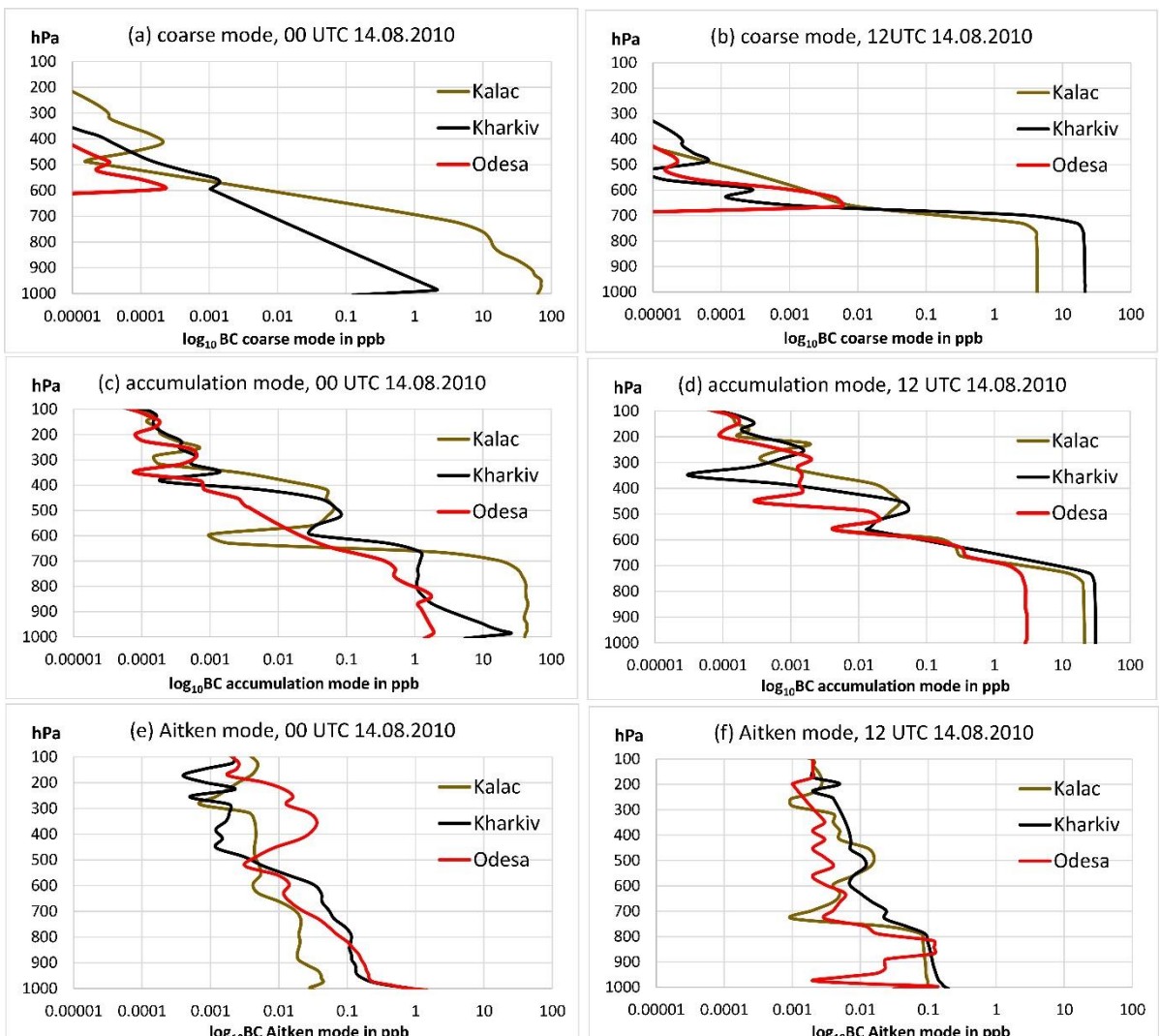

**Figure 5: The vertical profiles of BC for the coarse (a,b), accumulation (c,d) and Aitken (e,f) modes on examples of the Kalac, Kharkiv and Odesa sounding stations at 00UTC (a,c,e) and 12UTC (b,d,f) on 14th August 2010 (log₁₀ x-axis selected for better visibility and comparison).**

As seen in Fig. 5, BC was rather equally distributed in the 1000–700 hPa layer during daytime hours. But at nighttime, the BC was observed mostly in the boundary layer, and especially, in the coarse mode. The air temperature inversions and descending air during the night hours resulted in more intense deposition of coarse mode, and hence, making it difficult to detect coarse particles at the distances of more than 1000 km away from the active fires (represented by Odesa in Fig. 5). However, the accumulation mode could be transported through the atmosphere over such long distances and such transport was observed within the lowest 3-km layer.

Overall, it was found that the accumulation mode BC was a predominant size fraction in the region of study. While concentrations of the BC Aitken mode were small and the coarse mode deposited mainly close to the wildfires, the accumulation mode was transported by prevailing winds far from the burning areas, and mainly within the lowest 3-km layer (up to 700 hPa). In particular, this size fraction caused elevated BC concentrations over the region of study, with the highest concentrations near the surface at nighttime and during morning hours.

### 3.3 Ratio of aerosol compounds and observed direct aerosol effects during the wildfires

Aerosol impacts on the atmospheric composition during wildfire events depend on the intensity of the fire (Vadrevu et al., 2015), on the fuel (Christian et al., 2003; Lee et al., 2018) and the stage of the fire (e.g., open fire, smoldering fire (Lee et al., 2010; Popovicheva et al., 2019)). These features will influence, for example, the ratio of $CO/CO_2$, the amount of black carbon emissions as well as the vertical distribution of BC.

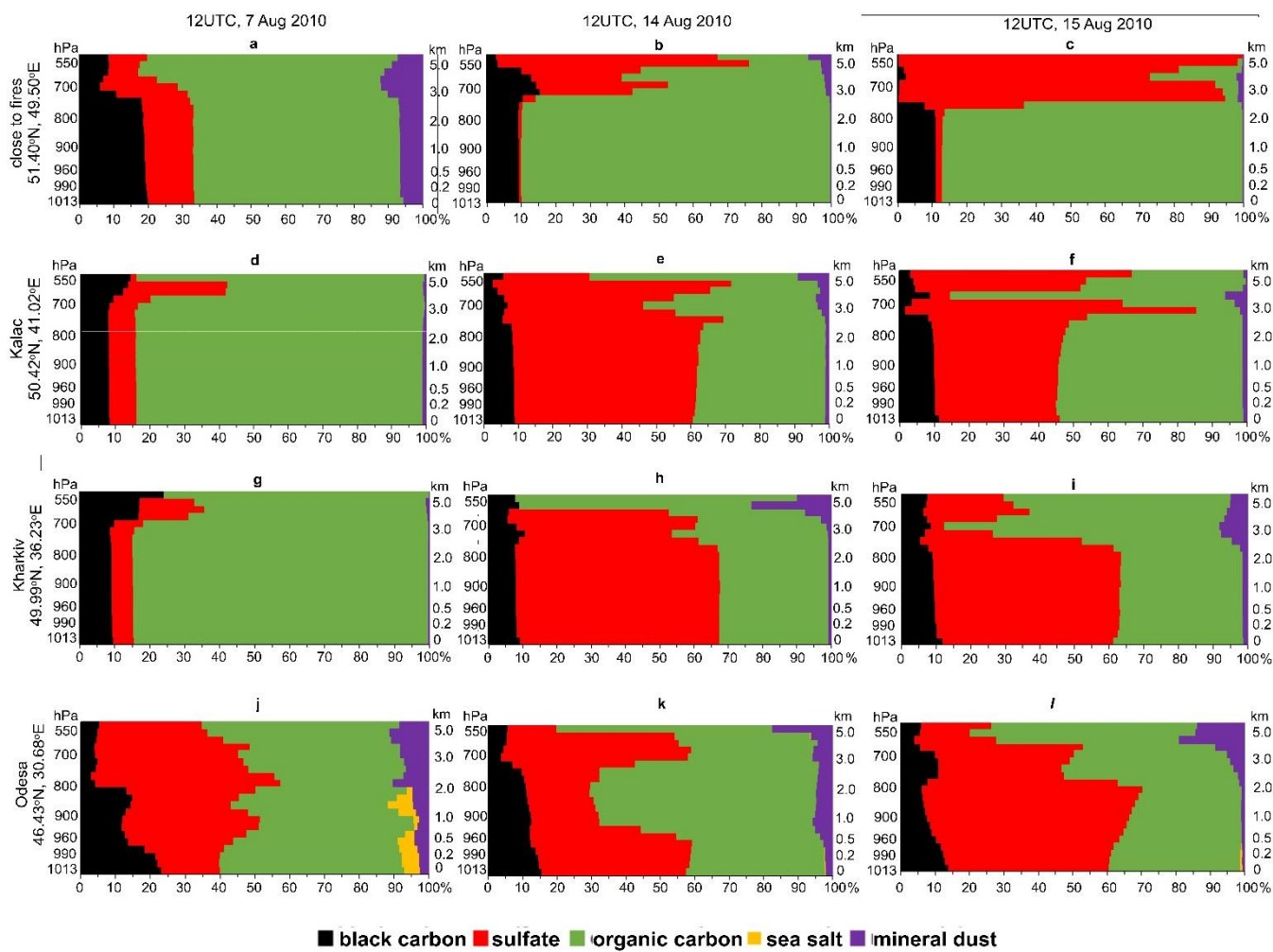

**Figure 6: Vertical distribution of ratio of aerosol compounds at different sites at 12 UTC on 7th August (a,d,g,j), 14th August (b,e,h,k) and 15th August (c,f,i,l) 2010.**


Enviro-HIRLAM simulations were conducted considering five main aerosol compounds: BC, organic carbon, sulfate, sea salt, and mineral dust. The highest values of BC-to-total aerosol content ratio were observed in the lower 200-m layer (up to 990 hPa) over fire areas, reaching 20% (see Fig.6a). BC normally accounts for 10% at the distance from wildfires by the prevailing wind (Fig. 6d-6i). The influence of local urban BC emissions is well seen on the example of Odesa (the most distant city among those presented), where the BC ratio increased to 15-24% near the surface (see Fig. 6j-6l). The vertical distribution of BC ratio was rather uniform from the surface to an altitude of 2 km (≈800 hPa).

Organic carbon is very often the main aerosol compound in the lower 2-5-km layer (≈800-550 hPa), especially near the fires, but also at the surface in general (e.g., Jimenez et al. 2009). Based on our results, sulfates prevailed throughout the lower and

middle troposphere at the distances, and at an altitude of over 3 km (≈700 hPa) near the wildfires. Hence, the aerosol effects over the region were mainly driven by organic carbon and sulfate aerosols. The largest BC contribution during the anticyclonic weather conditions seems to be in the boundary layer. In our analysis, the mineral dust ratio rarely exceeded 10%, whereas sea salt was detected in the atmosphere only near the sea (see Fig. 6j, 6l).

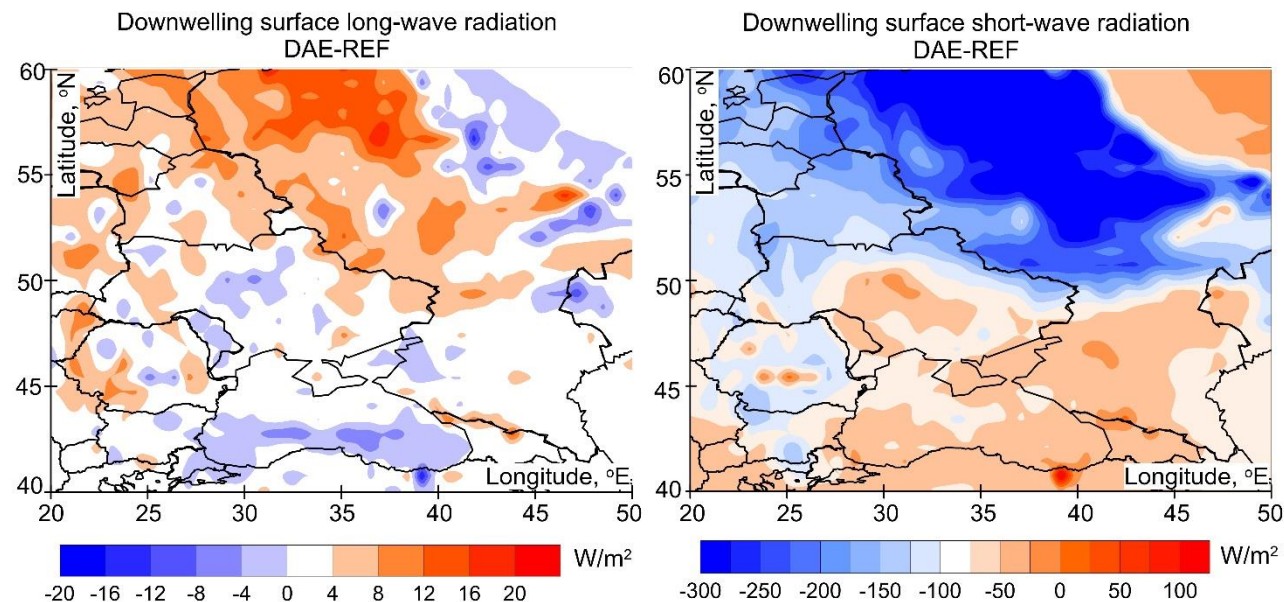

**Figure 7: Difference fields between the Enviro-HIRLAM DAE and REF model runs for downwelling surface long-wave and short-wave radiation at 12UTC on 14th August 2010.**

Based on our modelling results, continuous wildfires and aerosol emissions over the region as a consequence influenced the
radiative transfer via direct aerosol effects. The Enviro-HIRLAM DAE run showed both an increase and a decrease in downwelling surface long-wave and short-wave radiation. The most intense changes caused by direct aerosol effects were observed during the midday, reaching over ±20 W/m² for long-wave and over ±100 W/m² for short-wave radiation (see example in Fig. 7). The observed direct effects on the downwelling surface radiation co-aligned with the total aerosol impact. Our results indicate that when the BC concentration and the coarse mode exceeded 30 ppb and the BC-to-total aerosol mass
ratio was over 10% in the lower troposphere, the downwelling surface long-wave radiation tended to increase while the changes in short-wave radiation varied within -200…25 W/m².

In our simulations, the direct aerosol effects had an impact on temperature as well. The difference in the midday 2-m air temperature between DAE and REF runs varied from -3°C to 3°C (see Fig. 8). The 2-m air temperature decreased due to the
direct aerosol effects that prevailed during the wildfires. However, the 2-m air temperature was higher with the direct aerosol

effects in areas where DAE downwelling surface long-wave radiation was higher than 12 W/m$^2$ and the DAE-REF difference for surface short-wave radiation was not less than -150 W/m$^2$. Moreover, the 2-m air temperature was 1-4°C higher at the distance from the wildfires over the territories where the BC coarse mode exceeded mixing ratios of 20-30 ppb.

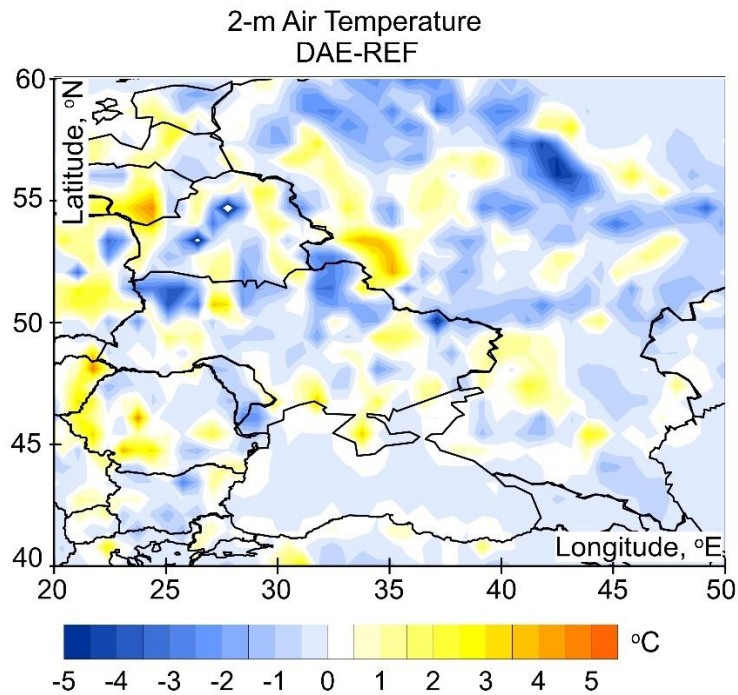


**Figure 8: Difference field between the Enviro-HIRLAM DAE and REF model runs for 2-m air temperature at 12UTC on 14$^{th}$ August 2010.**

### 3.4 Discussion

Wildfires typically produce large amount of BC emissions, which can be transported through the atmosphere far from the source (Eleftheriadis et al., 2009; Bond et al., 2013). During summer 2010, one of the most severe wildfires in Europe occured, enhanced by an unusually strong heat wave (Konovalov et al., 2011; Witte et al., 2011; Galytska et al., 2018). During observed stationary anticyclonic conditions, atmospheric horizontal and vertical transport of BC illustrated specific features. The territory of Ukraine, being among the most affected geographical regions (Galytska et al., 2016; Galytska et al., 2018), has
faced several elevated pollution episodes. The existing problems with national UA-AQMN made it impossible to perform an accurate analysis of different aerosol compounds, including BC distribution with Enviro-HIRLAM. Our results showed that the anticyclonic conditions caused the atmospheric transport of BC for more than 2000 km away from the burning areas. This elevated BC content was mainly formed by the accumulation mode, which is different to e.g., dust events (Kompalli et al., 2014), where coarse particles are the predominant size fraction. The Aitken mode contribution was much smaller in comparison

to the accumulation and coarse modes, and urban emissions frequently defined the spatial distribution of the Aitken mode BC. At the same time, the coarse mode BC was deposited not far from the burning areas, and it was rarely observed at distances of ca. 500 km away from the fires. Overall, during these particular wildfires, the area affected by intense BC deposition was rather large in comparison to other wildfires (Hodzic et al., 2007).

Based on our results, BC was mostly transported within the lowest 3-km layer (up to 700 hPa). At these altitudes, the BC ratio was 8-20% among all aerosol compounds. Frequent and strong air temperature inversions determined deposition during August 2010. Its influence during the stationary anticyclone was compared to that which low-level temperature inversions impact in areas with complex terrain on the example of Slovenia (Glojek et al., 2022). These conditions defined diurnal variations of BC and its vertical profiles. Despite the fact that the wildfires had the dominant role in the region, BC behaviour on a diurnal cycle

did not differ much in comparison to urban emissions (Sahu et al., 2011; Chen et al., 2014). The BC maxima near the surface occurred at nighttime and during morning hours, and minima were dependent on the boundary layer height development during the afternoon hours.

Being distributed in the lower 3-km layer, BC played a role in direct aerosol effects in the atmosphere. By using a fully online

coupled (integrated) model Enviro-HIRLAM, we estimated the direct aerosol effects on radiation fluxes and air temperature. The decrease in local radiation forcing (Chubarova et al., 2012) and a reduction in shortwave radiation (Pere et al., 2014) were observed after wildfire aerosol emissions during summer 2010. Previously, it was shown that the shortwave radiation is smaller by 70-84 W/m$^2$ in the diurnal averages (Pere et al., 2014). Our results confirmed the decrease in the downwelling surface short-wave radiation at the background as the result of the direct aerosol effects was up to -250 W/m$^2$ during the midday hours.

However, numerous regions were detected where the downwelling surface short-wave and long-wave radiation increased. The 2-m air temperature was higher despite considering the direct aerosol effects of the areas where the long-wave radiation increased by 20 W/m$^2$ and both BC coarse and accumulation modes exceeded values of 30 ppb. These hot-spots might be coaligned with the BC impacts at certain locations, as the BC influence is known for its localizing effects with no clear direct link between the pattern of forcing and the pattern of temperature change (Modak and Bala, 2019). Overall, regardless of the

emission source, the direct radiative forcing and the consequent air temperature increase was estimated for BC by different studies (Ma et al., 2018; Zhuang et al., 2019; Kostrykin et al., 2021) and cooling effects were also detected (Ma et al., 2018). In combination with other aerosol types, the effects on the radiative forcing are also variable (Kirkevag et al., 1999). Hence, not all areas with decreasing or increasing air temperature are directly related to the prevailing influence of certain aerosol compounds, but a net effect including physical, chemical, and meteorological impacts are being experienced.


In the absence of BC measurements, the ground-level monitoring network in Ukraine represents a challenge for the results' validation. Nevertheless, it gives an overall picture of the BC distribution during unfavourable weather conditions and makes it possible for the provision of assessments in Ukraine on the quantification of BC impact on human health and local

ecosystems in Ukraine as a whole. Moreover, the fact that wildfire emissions can cause local air temperature to increase amid elevated pollution levels is crucial for understanding the consequences for vulnerable people. This is especially relevant during heat wave events such as those in August 2010, which were discussed in this study.

**4 Conclusions**

Employing the Enviro-HIRLAM online integrated modelling system, the patterns of the spatio-temporal distribution of BC were estimated for a selected elevated pollution period of August 2010 in Ukraine, which resulted from severe forest fires in the central part of Russia. For the first time, spatio-temporal BC content, distribution of its different particle sizes, and BC ratio among other aerosol compounds in the atmosphere were analysed with the emphasis on extremely hot period. Moreover, for the first time, the seamless online integrated meteorology - atmospheric composition modelling approach (with the Enviro-HIRLAM model), compared to the classical off-line approach (separate meteorology and atmospheric chemical transport modelling), was applied to the geographical domain of Ukraine.

The highest BC content was observed in Ukraine in 2010 during two periods, 7-8 and 13-16 August, because of the prevailing air movement towards Ukraine from areas/ cells with burning forests. The stationary anticyclone and hence, favourable conditions with deep night-time air temperature inversions caused constant emission distribution and a large accumulative effect at the boundary level. BC was distributed over distances of more than 2000 km from the original emission sources. Over Ukraine, it reached 70-150 ppb for the accumulation mode and 50-80 ppb for the coarse mode. Anthropogenic emissions of BC Aitken mode prevailed over wildfires. Vertical transport was slower, and particles were mainly dispersed in the lowest 3-km layer. However, the fingerprint of BC coarse and accumulation modes could also be detected in the middle troposphere (i.e., up to 660-630 hPa level for the coarse mode and up to 590-450 hPa for the accumulation mode). The temperature inversions at night caused the diurnal variability of the BC vertical distribution. Here, concentrations descended to the lower layers at night and ascend at daytime by the weakening of downward air movement. For the near-surface level, the integral values over the Eastern Ukraine, Azov Sea, and Black Sea territories exceeded 800 and 150 ppb for the accumulation and coarse modes, respectively. The highest BC-to-total aerosol content ratio was observed in the lower 200-m layer (up to 990 hPa), reaching 10-24% near the fires and over urban areas. The dominant aerosol compounds were organic carbon and sulfates. Hence, direct aerosol effects on radiative and temperature regimes in the region were mostly typical for these compounds: a decrease in downwelling surface short-wave radiation and a decrease in 2-m air temperature. In areas with high BC content represented by accumulation and coarse mode, downwelling long-wave radiation with direct aerosol effects was higher by 20-25 W/m$^2$ during the midday hours. The 2-m air temperature was 1-4°C higher in these regions, with BC coarse mode exceeding 20-30 ppb.

The findings of this study are important and valuable for improving numerical weather prediction due to taking into account aerosol effects and for depicting the impacts of extreme events such as wildfires with atmospheric chemical transport modelling. The results are relevant for multi-scale assessment studies of atmospheric pollutants' impacts on population and ecosystem health, for climate adaptation and socio-economical related studies, for optimization and establishing of air quality monitoring stations and decision- and policy making processes.

Overall, the study needs to be further expanded in two directions. The first direction requires elaboration of ground-level BC measurements in Ukraine for modelling results validation and assessment of the negative BC impact on human health and local ecosystems in Ukraine with a future transition to mitigation measures and reduction strategy. The second direction should expand the analysis of direct and indirect aerosol effects by using seamless/ online-integrated modelling to distinguish the influences of different aerosol compounds.

**Appendix A**

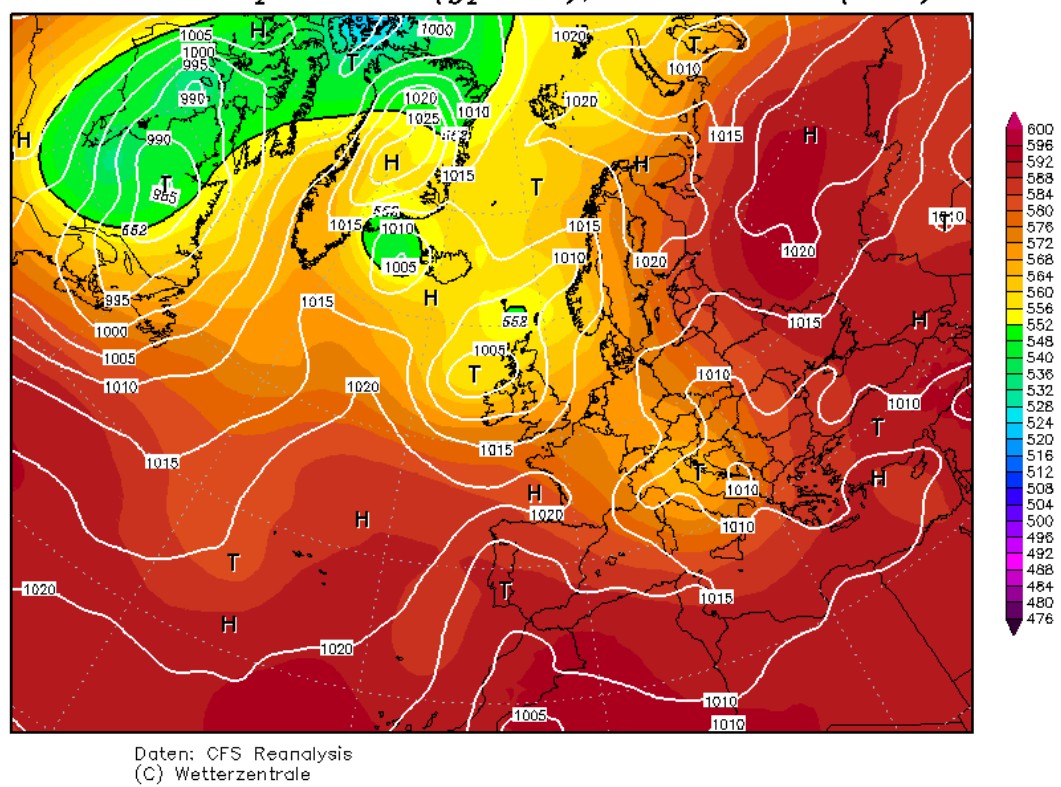

Figure A1. 500 hPa geopotential over Europe on 6th August 2010 (source: www.wetterzentrale.de)

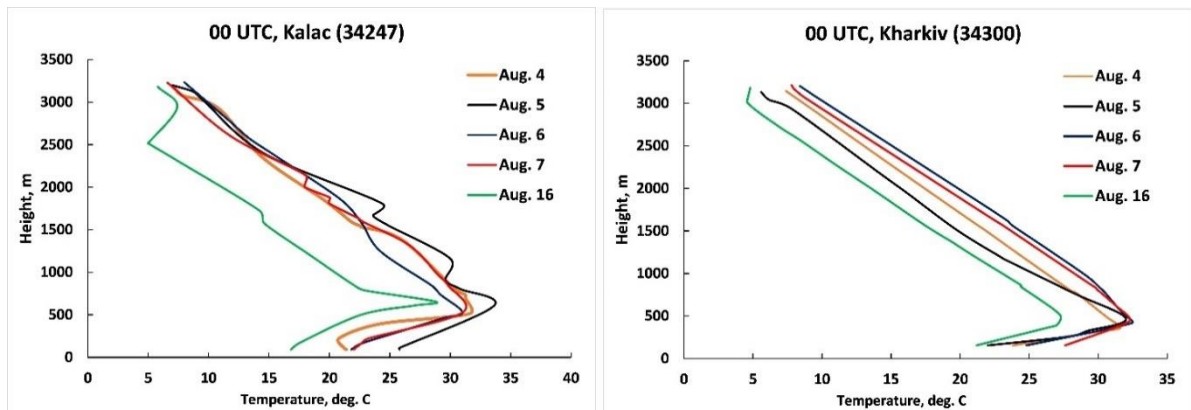

**Figure A2: The air temperature vertical profiles from the vertical sounding stations Kalac and Kharkiv for selected dates (4-7 and 16 August 2010) with the deepest surface temperature inversions at nighttime (00 UTC).**

*Code and data availability*. The code and data used in this study is available from the authors upon reasonable request.

*Author Contributions*. MS: Methodology, Investigation, Visualisation, Writing Original and Draft. LP: Investigation, Visualisation, Writing Original and Draft. SK: Conceptualization, Methodology, Supervision. AM: Conceptualization, Methodology, Supervision, Writing – Review and Editing. TP: Writing - Review and Editing. All authors provided comments on manuscript.

*Competing Interests*. The authors declare that they have no conflict of interests.

*Acknowledgements.* The study is part of the Enviro-PEEX on ECMWF (Pan-Eurasian EXperiment (PEEX; https://www.atm.helsinki.fi/peex) Modelling Platform research and development for online coupled integrated meteorology-chemistry-aerosols feedbacks and interactions in weather, climate and atmospheric composition multi-scale modelling) project (2018-2020).

The Enviro-HIRLAM model simulations were performed on the CSC (Center for Science Computing) Sisu HPC (Finland) during the Enviro-HIRLAM/ HARMONIE research training course at the Institute for Atmospheric and Earth System Research (INAR), University of Helsinki (UHEL).

The financial support was provided by the grant within ENVRIplus project for Multi-domain Access to RI platforms "The Influence of Land cover changes On Atmospheric Boundary Layer and Regional Climate Characteristics" (2018).

The work partially has been performed under the Project HPC-EUROPA3 (INFRAIA-2016-1-730897), with the support of the EC Research Innovation Action under the H2020 Programme; in particular, the authors gratefully acknowledge the computer resources and technical support provided by Center for Science Computing (CSC) HPC (Finland).

This study was carrying out by the financing of Ukrainian Hydrometeorological Institute within the framework of the State Emergency Service of Ukraine and National Academy of Sciences of Ukraine.

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
