# Peer review of "Enviro-HIRLAM model estimates of elevated black carbon pollution over Ukraine resulted from forest fires"

_Atmospheric Chemistry and Physics, 2022_

## Author Comment (AC2)

**Answer to Reviewer 2**

Dear Anonymous Reviewer,

We are thankful for your constructive criticism and comments to the manuscript "Estimation of elevated black carbon episode over Ukraine using Enviro-HIRLAM" submitted to the ACP journal. Please, see below our replies and changes/ modifications to the original manuscript.

*R2. The authors estimated the elevated black carbon (BC) levels in temporal and three-dimensional spatial scales after the forest fire events occurred in summer 2010 at the center of the European territory of Russia. Throughout the manuscript, the writing, however, appears somewhat rambling, especially for the 'Results and discussion" section. The authors should give more discussion and interpretation for their results.*

A: The Section "Results and Discussion" was revised and re-written accordingly. We expand our analyses with results by adding the analyses of aerosol component ratio, direct aerosol effects on downwelling surface short / long-wave radiation fluxes and air temperature. There was added Aitken mode for previously presented results.

[revised manuscript text omitted]

Moreover, the sections 3.1 and 3.2 were also improved by addition more detailed analysis for the Aitken mode (including updated Figures 2-3-4), and in particular, by adding the following text:

For sub-section 3.1 (lines: 176-179; 203-211; 216-228):

"At the beginning of August 2010 before the wildfire episode, BC content in Ukraine was low and mostly represented by the accumulation mode with the values of 1-8 ppb. The highest BC concentrations reached 16 ppb (accumulation mode) and was observed at the near-surface level near small local wildfires in the central part of Ukraine. BC Aitken and coarse modes did not exceed 0.5 ppb."

&

"In general, the wildfire emissions have large accumulative effect in the near-surface layer. Total accumulated amount of BC for the period 3–18 August 2010 reached 13500 ppb (for accumulation mode) and 2200 ppb (coarse mode) in the lower tropospheric layer near the burning areas/ cells (Fig. 3). Total accumulated amount of BC Aitken mode caused by wildfires was about 15-30 ppb, whereas the maximum accumulated effects was observed in cities and caused by local anthropogenic emissions (see Aitken mode on Fig. 3). A large amount of combustion products was transported through the atmosphere to the south-west and deposited over territories of the Eastern Ukraine, the Azov and Black Seas. The integral values of BC on these territories exceeded 800 and 150 ppb for accumulation and coarse modes, respectively. This is well seen from Fig. 3 where the regions were affected by intensive deposition processes. Due to smaller sizes of the particles, the accumulation mode has larger spatial coverage and more smooth distribution than the coarse mode."

&

"For the studied period, in general, ground-based dust measurements also showed elevated levels connected with forest fires. Almost all geographical regions except northern part of Ukraine have clear maxima of ground-based dust concentrations during August 2010. The highest exceeding over urban background values in August were observed on the eastern territories. Here, the integral value of the near-surface BC concentrations for the coarse mode was higher than 250 ppb. For the cities on the east, the dust concentrations were 27–47% higher than average dust content in 2010. Moreover, these were also 23–72% higher than multi-year average concentrations for month of August. Overall dust content on the east was 0.02–0.07 $\mu g/m^3$ higher than usually in August.

Large difference in dust content between August and other months was observed on the seashore of the Azov Sea and in the central part of Ukraine. The concentrations were higher on 0.05–0.25 $\mu g/m^3$ than usually in the same month. Majority of cities in the central part of Ukraine showed 17–73% higher dust concentrations than average in 2010 and 8–45% higher than usually in August. In the western parts of Ukraine, the integral values were lower than 100 and 500 ppb for coarse and accumulation modes, respectively (Fig. 3)."

for sub-section 3.2 (lines: 254-258):

"In contrast to the BC coarse and accumulation modes, Aitken mode was observed throughout the entire troposphere up to 200 hPa level (Fig. 4) height, however the wildfire emissions did not prevail over the anthropogenic emissions. The BC Aitken mode rarely exceeded 0.2 ppb at 950 hPa level. Therefore, concentrations more than 1 ppb were observed only near the surface and had clear maximum over urban areas (see Fig. 3). This is the reason why the BC Aitken mode near the surface was higher in more distant Odesa than in Kalac region"

& sub-section 3.4 "Discussion" was added (lines: 316-344):

"BC dispersion and vertical distribution during stationary anticyclonic conditions had their own specific features. The most typical are uniform vertical profile up to 700 hPa (≈3 km) and BC ratio of 8-20% among all aerosol compounds. Very often there is no clear maximum within lowest 3-km layer especially for coarse and accumulation mode. Urban areas with additional anthropogenic emissions are the exception where maximum formed near the surface. This distribution differs from other synoptic conditions or its averaging, which results in maximum BC concentration outside the emission areas at an altitude of 1.5 km (Kostrykin et al., 2021).

The decrease of radiation forcing (Chubarova et al., 2012) and reduction of shortwave radiation (Pere et al, 2014) were observed after wildfire aerosol emissions during summer 2010. Previously it was shown that shortwave radiation is smaller up to 70-84 $W/m^2$ in diurnal averages (Pere et al., 2014). Our results confirmed the decrease of downwelling surface short-wave radiation at the background as the result of direct aerosol effects up to -250 $W/m^2$ during the midday hours. However, numerous spots were observed where downwelling surface short-wave and long-wave radiation increased. 2-m air temperature is higher considering direct aerosol effects in areas where long-wave radiation increased up to 20 $W/m^2$ and both BC coarse and accumulation mode exceeded 30 ppb. These increasing might correspond to BC impact at certain locations, as BC influence is known by its localizing effects with no clear directs link between the pattern of forcing and pattern of temperature change (Modak and Bala, 2019). Overall, direct radiative forcing and air temperature increase was estimated for BC regardless emission source (Ma et al., 2018; Zhuang et al., 2019; Kostrykin et al., 2021); however cooling effects also was detected (Ma et al., 2018). In combination with other aerosol types the effects on radiative forcing also is different (Kirkevag et. al, 1999). Hence, not all areas with air temperature increasing or decreasing could be directly connected with the prevailing influence of certain aerosol compound.

The absence of BC measurements at ground-based monitoring network in Ukraine represents a challenge for results validation. Nevertheless, it gives overall picture on BC distribution during unfavorable weather conditions and makes possible to provide assessments in Ukraine on how BC influenced human health and local ecosystems. Moreover, the fact that wildfire emissions could cause local air temperature increasing amid elevated pollution levels is crucial for understanding the consequences for vulnerable people. This is especially relevant during the heat wave events likewise it was in August 2010 discussed in this study.
"

*R2. The authors need to explain exactly what the novel insights of their work are, and how their finding are relevant to atmospheric chemistry and radiative transfer.*

A: The Abstract, sections 3 "Results and Discussion" and "Conclusions" were revised and improved by including key findings of this study, and in particular, this study in Abstract:

"BC contribution was found to be 10-20% among total aerosol compounds near the wildfires in the lower 2-km layer. At the distance, BC exceeded 10% only over the urban areas. In the areas with high BC content represented by both accumulation and coarse mode, downwelling surface long-wave radiation increased up to 20 $W/m^2$ and 2-m air temperature increased on 1-4°C during the midday hours." (Lines: 17-20).

In addition, "This study is the first for Ukraine which describe elevated spatio-temporal BC content in the three-dimensional scale, distribution of its different particle sizes and BC ratio among other aerosol compounds in the atmosphere with the emphasize on extremely hot period." (Lines: 79-82).

Moreover, for the first time, the seamless online integrated meteorology - atmospheric composition modelling approach (Enviro-HIRLAM model), compared to classical off-line approach (separate meteorology and atmospheric chemical transport modelling), was applied to geographical domain of Ukraine.

*R2. Especially compared with the earlier works that investigated the forest fire events occurred in summer 2010, the authors could state what questions had been left open, and what the present paper now addresses.*

A: The section "Introduction" was updated by including the following relevant publications analysis: "In contrast to other the aerosol compounds, BC typically cause radiative a positive forcing (Bond et al., 2013; Stjern et al., 2017) which intensity depends on the particle size (Matsui et al., 2018). Consequently, the heating effect is generally observed from wildfire emissions (Kostrykin et al., 2021), and also from anthropogenic sources (Zhuang et al., 2019). Sometimes a cooling effect was also detected (Ma et al., 2018). All these effects, however, are localized (Modak and Bala, 2019)." (Lines: 34-38)

&

"Described wildfire events frequently occur in Ukraine or in territories of neighboring countries that develop into elevated pollution events. Unfortunately, the existing official Ukrainian air quality monitoring network (UA-AQMN) does not provide any measurements of BC. Hence, the modelling remains the only possible way to estimate the spatio-temporal variability of BC in Ukraine, and to explore the consequences of elevated pollution episodes and assess the short-term impacts in the region. The lack of observation capacity on BC on a national level caused a gap in knowledge on how BC is distributed; what are the impacts on local ecosystems; and which mitigation measures are needed to improve the situation. This study is the first in Ukraine which describes the elevated spatio-temporal BC content on the three-dimensional scale, explores the distribution of BC in particle sizes and compares the BC ratio between other aerosol components in the atmosphere with the emphasize on extremely hot weather episodes. Furthermore, we aimed to estimate the impact of wildfire emissions on the surface by considering direct aerosol effects, as the response of radiative and temperature regimes varying in different regions and depending on the ratio of aerosol components." (Lines: 74-84)

&

additionally, analyzed in this study "Direct aerosol effects on downwelling surface short-wave/ long-wave radiation and 2-m air temperature also are discussed." (Lines: 95-96)

&

in sub-section 3.4 "Discussion"

"

The BC dispersion and vertical distribution during stationary anticyclonic conditions had their own specific features. The most typical cases were a uniform vertical profile up to 700 hPa ($\approx$3 km) with a BC ratio of 8-20% among all the aerosol compounds. Very often there was no clear maximum within the lowest 3-km layer especially for the coarse and accumulation modes. The urban areas with additional anthropogenic emissions were an exception where a maximum formed near the surface. This distribution differed from the other synoptic conditions, which resulted in a maximum BC concentration outside the emission areas at an altitude of 1.5 km (Kostrykin et al., 2021).

The decrease of radiation forcing (Chubarova et al., 2012) and a reduction of shortwave radiation (Pere et al, 2014) were observed after wildfire aerosol emissions during summer 2010. Previously it was shown that the shortwave radiation is smaller on 70-84 W/m$^2$ in the diurnal averages (Pere et al., 2014). Our results confirmed the decrease in the downwelling surface short-wave radiation at the background as the result of the direct aerosol effects were up to -250 W/m$^2$ during the midday hours. However, numerous spots were observed, where the downwelling surface short-wave and long-wave radiation increased. The 2-m air temperature was higher in spite of considering the direct aerosol effects of the areas where the long-wave radiation increased up to 20 W/m$^2$ and both BC coarse and accumulation mode exceeded values of 30 ppb. These hot-spots might coaligned with the BC impacts at certain locations, as the BC influence is known by its localizing effects with no clear directs link between the pattern of forcing and pattern of temperature change (Modak and Bala, 2019). Overall, the direct radiative forcing and the consequent air temperature increase was estimated for BC regardless of the emission source (Ma et al., 2018; Zhuang et al., 2019; Kostrykin et al., 2021); however cooling effects also was detected (Ma et al., 2018). In combination with other aerosol types the effects on the radiative forcing also is variable (Kirkevag et. al, 1999). Hence, not all areas with air temperature increasing or decreasing are not directly connected with the prevailing influence of certain aerosol compound but a net effect including physical, chemical and meteorological effects are being experienced.

In the absence of BC measurements, ground-based monitoring network in Ukraine represents a challenge for the results validation. Nevertheless, it gives an overall picture of the BC distribution during unfavorable weather conditions and makes it possible for a provision of assessments in Ukraine on the quantify of BC influenced human health and local ecosystems in Ukraine as a whole. Moreover, the fact that the wildfire emissions can cause local the air temperature to increase amid elevated pollution levels is crucial for the understanding the consequences for vulnerable people. This is especially relevant during the heat wave events such as in August 2010, which is discussed in this study." (Lines: 318-344)

& in Section "Conclusions":

"Overall, the study needs further development in two directions. First direction requires elaboration of ground-based BC measurements in Ukraine for modelling data validation and assessment the negative BC impact on human health and local ecosystems in Ukraine with future transition to mitigation measures and reduction strategy. Second direction should expand the analysis of direct and indirect aerosol effects using online-integrated modelling with separating the influences of different aerosol compounds." (Lines: 382–386)

***General Comments*:**

*R2. The authors claimed that the work focused on the horizontal and vertical variability of BC concentrations over Ukraine during a wildfire episode in August 2010. However, the author just*

*presented the BC concentrations after the forest fire evens, not comparing with BC levels before the wildfire evens; how to demonstrate that the BC levels were elevated?*

A: Information about BC concentrations before wildfire episode was added to the Section 3.1 "Synoptic weather situation and the dispersion of wildfire emissions during August 2010 in Ukraine". The following text was added (Lines: 176-179):

"At the beginning of August 2010 before the wildfire episode, the BC content over Ukraine was low and mostly it was represented by the accumulation mode with the values of 1-8 ppb. The highest BC concentrations reached 16 ppb (accumulation mode) and was observed at the near-surface level near small local wildfires in the central part of Ukraine. The BC concentrations in the Aitken and coarse modes did not exceed 0.5 ppb."

*R2. Additionally, the authors should valid their simulations of BC concentrations. Could they provide some BC measurements rather than dust observations to verify their simulations and discussed the uncertainties of the simulations.*

A: Unfortunately, this is a big and serious problem for Ukraine. There are no monitoring sites which can provide black carbon (BC) measurements (and also many others chemical species). Hence, the modelling remains the only available tool for the Ukrainian territory, which allowed to estimate the spatio-temporal distribution of such species like BC, organic carbon, etc. As a result, there is a gap in scientific knowledge about BC in Ukraine, and there are no assessments how it is distributed; how it influences local ecosystems, human health; and, consequently, no relevant mitigation measures were taken.

Moreover, the existing official Ukrainian air quality monitoring network (UA-AQMN) was developed and established in the former USSR, and it was not significantly improved or upgraded during the last decades. The monitoring sites (more than 120 in total) of the network do not measure PM; and dust – is the only pollutant (measured by official air quality stations) which could describe aerosol content. We add details on it in the "Methods"-section, including what "dust" is and meaning according to existing local regulations in Ukraine. Of course, the process for EU Directives implementation has been already started in Ukraine, but it is still in slow progress and not finished yet.

As soon as BC measurements will be launched as part of UA-AQMN, it would be possible to study it accurately and precisely. But nowadays only the modelling approach could be used. Nevertheless, first assessments are necessary for interests and needs of Ukrainian national economy following UN Sustainable Development Goals, and in particular, SGDs 3, 9, 13.

We described and explained this in the revised version of the manuscript.

Accordingly, the following changes/additions were incorporated in the revised manuscript:

in Section "Introduction" (lines: 74-84):

"Described wildfire events frequently occur in Ukraine or in territories of neighboring countries that develop into elevated pollution events. Unfortunately, the existing official Ukrainian air quality monitoring network (UA-AQMN) does not provide any measurements of BC. Hence, the modelling remains the only possible way to estimate the spatio-temporal variability of BC in Ukraine, and to explore the consequences of elevated pollution episodes and assess the short-term impacts in the region. The lack of observation capacity on BC on a national level caused a gap in knowledge on how BC is distributed; what are the impacts on local ecosystems; and which mitigation measures are needed to improve the situation. This study is the first in Ukraine which describes the elevated spatio-temporal BC content on the three-dimensional scale, explores the distribution of BC in particle sizes and compares the BC ratio between other aerosol components in the atmosphere with the emphasize

on extremely hot weather episodes. Furthermore, we aimed to estimate the impact of wildfire emissions on the surface by considering direct aerosol effects, as the response of radiative and temperature regimes varying in different regions and depending on the ratio of aerosol components."

& in sub-section 2.2 "Additional data for analysis" (lines: 145-148)

"Unfortunately, UA-AQMN does not provide measurements of BC, moreover PM also is not measured. Dust – is the only available pollutant which could be compared to the modeled aerosol species. The dust is measured at more than120 monitoring sites, and measurements contain all coarse aerosol particles regardless of their origin (Nadtochii et al., 2019). Dust concentration on these sites is measured using the method of weighing the total suspended particles (RD, 1991)."

*R2. The authors claimed that the features of BC distributions that they presented in this work not only resulted from biomass burning, but also were affected by the local sources of fires. Could they estimate how much the influence?*

A: We accurately evaluated information about sources of local fires for studied period in Ukraine and hence, re-evaluated results and excluded Fig. 7.

***Specific comments*:**

*R2. Overall, the paper is rather poorly written with many grammar/ formatting mistakes.*
A: The manuscript was revised following remarks/comments, portions of the text were re-written, grammar were re-checked and we hope that the manuscript text improved.

*R2. Some simple things, such as the superscripts and subscripts of "PM10" "PM2.5" and "μg/m3" (several times),*
A: Superscripts and subscripts were checked and corrected accordingly.

*R2. inconsistent format of the figure legends (font size and line space (caring Figure 1), font bold (caring Figure 2)).*
A: Figures' legends were checked and corrected accordingly. Moreover, all illustrations/ figures were re-drawn and higher resolution and clarity.

*R2. These mistakes indicate that proper care was not taken to proofread the paper prior to submission.*
A: Mistakes were corrected following remarks/comments, some text/paragraphs of the manuscript were re-written, grammar re-checked, additional proof-reading was also performed. By doing this, we hope that the manuscript text became of better quality.

*R2. Figure 2 is from the Climate Forecast System (CFS) Reanalysis (source: www.wetterzentrale.de). It can be shown in the Supporting Information.*
A: Figure 2 moved to Appendix A.

*R2. The "3.1 Synoptic weather situation during summer 2010 in Ukraine" just has one paragraph, which can be combined with the "3.2 Dispersion of wildfire emissions".*

A: Section 3.1 was combined with 3.2 and 3.4. We entitled the Section "3.1 Synoptic weather situation and the dispersion of wildfire emissions during August 2010 in Ukraine". Information about BC content before wildfire event was also added to this Section.

*R2. Page 7 /Line 160-164: The authors discussed the horizontal distribution of BC concentrations not only in the accumulation mode but also in the coarse mode. However, the authors only showed the BC distribution in the accumulation mode in Figure 3. Why not present the simulations of coarse mode.*

A: The BC distribution for Aitken and coarse modes was added in Figure 2 (Fig. 3 in previous version of the manuscript).

And additionally, the following new text included "The highest values for BC Aitken mode did not exceed 6 ppb near the fires and the simulated values were lower than 2 ppb at distances from the burning areas. Moreover, the highest values of Aitken mode were intermittently observed over urban areas reaching 10 ppb." (Lines: 193-195).

*R2. Page 9 /Line 195: The authors highlighted that the elevated BC concentrations were detected at 590 hPa (for the coarse mode) and at 550 hPa (for the accumulation mode). However, Figure 5 shows that the BC concentrations were very low with values near 0 in the <650 hPa layers. Could the authors clear how they identify the elevated BC concentrations at 590 hPa (for the coarse mode) and at 550 hPa (for the accumulation mode).*

A: We improved our Figure 4 (Fig. 5 in previous version of the manuscript), Aitken mode also included. Now, the heights become visible, where BC was detected during the wildfire event. Additionally, the following new text was added:

"In contrast to BC coarse and accumulation mode, Aitken mode was observed throughout the entire troposphere up to 200 hPa level (Fig. 4), however wildfires did not prevail over anthropogenic emissions. BC Aitken mode rarely exceeded 0.2 ppb over 950 hPa level. Concentrations more than 1 ppb were observed only near the surface and had clear maximum over urban areas (see Fig. 3). This is the reason why BC Aitken mode near the surface is higher in more distant Odesa than in Kalac." (Lines: 254-258)

*R2. Page 10 /Line 219-225: This paragraph introduced how to process the ground-based dust measurements. It should be in the "Data and methods" rather than the "Results and discussion".*

A: We re-wrote original text and moved to section 2 "Data and methods". In particular, the revised text is the following:

"Unfortunately, UA-AQMN does not provide measurements of BC, moreover PM also is not measured. Dust – is the only available pollutant which could be compared to aerosol species. Dust is measured at more than 120 monitoring sites, and measurements contain all coarse aerosol particles regardless of their origin (Nadtochii et al., 2019). Dust on these sites is measured using the method of weighing the coarse aerosol particles (RD, 1991) with the size not less than $PM_{10}$ fraction." (Lines: 145-148)

*R2. Page 11 /Line 239-241: The two sentences are confusing and needs to be reworded.*

A: Confusing sentences were removed/ rewritten. See our answer on your question in your next question and our answer to that.

*R2. Page 12 /Line 255-259: The authors discussed the "tails" of the BC distribution in the coarse mode, attributing to atmospheric transport and the impact of local fires. Why the BC distribution in accumulation mode did not present "trail" feature. How the effects of the atmospheric transport and local fires on the BC distribution in the accumulation mode?*

A: We change the color scale of the BC accumulation mode in Fig. 3 (fig 5 in the previous version of the manuscript). Also, we reconsider the results and removed the explanation about the possible impact of local fires. We combined this Section with Section 3.1. In particular, the revised figure is now as follows:

[Figure]

**Figure 3: The integral value of the near-surface BC concentrations for the Aitken, accumulation and coarse modes for the period 3-18 August 2010.**

& the revised text is the following (Lines: 203-228):

"In general, the wildfire emissions have a large accumulative effect in the near-surface layer. Total accumulated amount of BC for the period 3–18 August 2010 reached 13500 ppb (for the accumulation mode) and 2200 ppb (for the coarse mode) in the lower tropospheric layer near the burning areas (Fig. 3). The total accumulated amount of BC Aitken mode caused by wildfires was about 15-30 ppb, whereas the maximum accumulated impacts were observed in the cities (see e.g., Aitken mode on Fig. 3). A large amount of combustion products was transported through the atmosphere to the south-west and deposited over the territories of the Eastern Ukraine, the Azov and Black Seas. The integral values of BC on these territories exceeded values of 800 and 150 ppb for the accumulation and for the coarse modes, respectively. This is well seen in Fig. 3 where the regions were affected by intensive deposition processes. Due to the smaller sizes of the particles, the accumulation mode had a larger spatial extent and more smooth distribution than the coarse mode.

For the studied period, in general, the ground-based dust measurements also showed elevated levels connected with the forest fires. Almost all the geographical regions except northern part of Ukraine experienced a clear maxima of ground-based dust concentrations during August 2010. The highest exceeding over the urban background values in August were observed on the eastern territories. In this case, the integral value of the near-surface BC concentrations for the coarse mode was higher than 250 ppb. For the cities on the east, the dust concentrations were 27–47% higher than the average dust content in 2010. Moreover, these values were also 23–72% higher than multi-year average concentrations for the month of August. Overall, the dust content on the east Ukraine was 0.02–0.07 $\mu g/m^3$, which was higher than usually occurred in August.

Large difference in the dust content between August and other months was observed on the seashore of the Azov Sea and in the central part of Ukraine. The concentrations were higher by 0.05–0.25

µg/m³ than usually in the same month. A majority of cities in the central part of Ukraine showed 17–73% higher dust concentrations than average in 2010 and concurrently 8–45% higher than usually in August. In the western parts of Ukraine, the integral values were lower than 100 and 500 ppb for coarse and accumulation modes, respectively (Fig. 3)."

*R2. Conclusions: The authors need to clear the novel insights of their work and the significance of their finding.*

A: The section 4 "Conclusion" was re-written to underline more details on findings of this study, by including the following text:

"The highest BC-to-total aerosol content ratio was observed in the lower 200-m layer (up to 990 hPa) reaching 10-24% near the fires and over urban areas. The dominant aerosol compounds were organic carbon and sulfates. Hence, direct aerosol effects on radiative and temperature regimes in the region mostly were typical for these compounds: decreasing in downwelling surface short-wave radiation and decreasing of 2-m air temperature. In areas with high BC content representing by accumulation and coarse mode, downwelling long-wave radiation with direct aerosol effects was higher up to 20-25 W/m² during the midday hours. 2-m air temperature was 1-4°C higher in these regions with BC coarse mode exceeded 20-30 ppb." (Lines: 368-374).

Additional text was also included "For the first time, spatio-temporal BC content, distribution of its different particle sizes, and BC ratio among other aerosol compounds in the atmosphere with the emphasize on extremely hot period were analyzed." (Lines: 349-350)

"Moreover, for the first time, the seamless online integrated meteorology - atmospheric composition modelling approach (with Enviro-HIRLAM model), compared to classical off-line approach (separate meteorology and atmospheric chemical transport modelling), was applied to geographical domain of Ukraine." (Lines: 350-354)

Plus, extra text added was added to the paper as follows:

"The obtained results of the study are important and significant for improvement of quality of numerical weather prediction and for depicting the impacts of the extreme events with atmospheric chemical transport modelling. The findings are relevant for assessment studies of atmospheric pollutants impact on population and ecosystems health, for climate adaptation and socio-economical related studies, for optimization and establishing of air quality monitoring stations, decision- and policy making process, etc." (Lines: 376-380)

---

## Author Comment (AC3)

**Answer to Reviewer 1**

**Dear Anonymous Reviewer,**

We are thankful for your constructive criticism and comments to the manuscript "Estimation of elevated black carbon episode over Ukraine using Enviro-HIRLAM" submitted to the ACP journal. Please, see below our replies and changes/ modifications to the original manuscript.

R1. This is a modelling study about black carbon particles but without any validation from measurements, at least some ground measurements of BC are required.

A: We totally agree with the comments on validation and included and clarified more details in the text concerning that aspect. Unfortunately, validation data is a big and serious problem for Ukraine. There are no monitoring sites that provide black carbon (BC) measurements and this is valid for a wide variety of chemical species. Hence, the modelling remains the only available tool for the Ukrainian territory, which allowed to estimate the spatio-temporal distribution of species like BC, organic carbon, etc. As a result, there is a gap in scientific knowledge about BC in Ukraine, and there are no assessments how BC is distributed; how it influences local ecosystems, human health; and, consequently, no relevant mitigation measures were taken.

Moreover, the existing official Ukrainian air quality monitoring network (UA-AQMN) was developed and established in the former USSR, and it was not significantly improved or upgraded during the last decades. The monitoring sites (more than 120 in total) of the network do not measure PM; and dust – is the only pollutant (measured by official air quality stations) which could describe aerosol content and concentration. To note the reader on this, we added details on it in the "Methods"-section, including the definition of "dust" according to the existing local regulations in Ukraine. Of course, the process for EU Directives implementation has been already started in Ukraine, but it is still in slow progress and not finished yet.

As soon as BC measurements will be launched as part of UA-AQMN, it would be possible to study the spatio-temporal variability of BC more accurately and precisely. But nowadays only the modelling approach can be used. Nevertheless, the first assessment such as the current work, are necessary to raise awareness for interests and needs of Ukrainian national economy following UN towards Sustainable Development Goals, and in particular, SGDs 3, 9, 13.

In practice, we described and explained this in the revised version of the manuscript.

Accordingly, the following changes/additions were incorporated in the revised manuscript: in Section "Introduction":

"Described wildfire events frequently occur in Ukraine or in territories of neighboring countries that develop into elevated pollution events. Unfortunately, the existing official Ukrainian air quality monitoring network (UA-AQMN) does not provide any measurements of BC. Hence, the modelling remains the only possible way to estimate the spatio-temporal variability of BC in Ukraine, and to explore the consequences of elevated pollution episodes and assess the short-term impacts in the region. The lack of observation capacity on BC on a national level caused a gap in knowledge on how BC is distributed; what are the impacts on local ecosystems; and which mitigation measures are needed to improve the situation. This study is the first in Ukraine which describes the elevated spatio-temporal BC content on the three-dimensional scale, explores the distribution of BC in particle sizes and compares the BC ratio between other aerosol components in the atmosphere with the emphasize on extremely hot weather episodes. Furthermore, we aimed to estimate the impact of wildfire

emissions on the surface by considering direct aerosol effects, as the response of radiative and temperature regimes varying in different regions and depending on the ratio of aerosol components."

**& in sub-section 2.2 "Additional data for analysis"**

"Unfortunately, UA-AQMN does not provide measurements of BC, moreover PM also is not measured. Dust – is the only available pollutant which could be compared to the modeled aerosol species. The dust is measured at more than120 monitoring sites, and measurements contain all coarse aerosol particles regardless of their origin (Nadtochii et al., 2019). Dust concentration on these sites is measured using the method of weighing the total suspended particles (RD, 1991).

**R1. The modelling setup is too rough.**

A: The section 2.1 "Enviro-HIRLAM setup" was updated with including more details. The following text was added for meteorological and atmospheric composition input for the model runs:

"The initial and boundary conditions (ICs/BCs) for meteorology include components of winds speed, air temperature and specific humidity extracted at all model levels and at every 3-hour interval from ERA5 model archives at the European Centre for Medium-Range Weather Forecasts (ECMWF). The sea surface temperature and the conventional BUFR (The Binary Universal Form for the Representation of meteorological data) observations (for data assimilation) were extracted at every 12-hour and 3-hour interval, respectively. The ICs/BCs for gas and aerosol concentrations included 3D fields of mixing ratio of aerosol components (dust, hydrophilic and hydrophobic organic matter and black carbon, sulphates) as well as gases (O3, SO2, NO2, NO, hydrogen peroxide (H2O2), hydroxyl radical (OH), nitrate radical (NO3), hydroperoxyl radical (HO2), dimethyl sulfide (DMS)) were extracted at every 3-hour interval from Copernicus Atmosphere Monitoring Service (CAMS) of ECWMF."

The following text was added for emission inventories used for the model runs:

"A suite of emission inventories (EIs) was utilized in the model runs, including anthropogenic and biomass burning (wildfires). In more detail, the EIs (given in geographical latitude/longitude domain) – for the global biomass burning (wildfires) emissions (IS4FIRES; Sofiev et al., 2012), Evaluating the Climate and Air Quality Impacts of Short-Lived Pollutants (IIASA's ECLIPSE, 5th version) (gridded emissions of gases and aerosols of SO2, NOx, NH3, nmVOC, BC, OC, PM2.5, PM10, CO, CH4 for SNAP codes (Selected Nomenclature for Air Pollution) such as industry, transportation, agriculture, etc.), shipping emissions (include SO2, BC and OC), emissions of DMS (Nightingale et al., 2000). The mission preprocessor includes both vertical and temporal profiles for the model setup."

*R1.* Not just that, but all the presentation throughout the texts shows the authors have not really performed any meaningful data analysis.

A: We expand our analyses with results by adding the analyses of aerosol component ratio, direct aerosol effects on downwelling surface radiation fluxes and air temperature. There was added Aitken mode for previously presented results. Based on this work, the section "Results and Discussions" was revised.

The section 3 "Results and Discussion" was updated with including more details.

In particular, new sub-section "3.3 Ratio of aerosol compounds and observed direct aerosol effects during the wildfire episode" was included with Figures 5-6-7.

"Aerosol impacts on the atmospheric composition during wildfire events depend on the intensity of the fire (Vadrevu et al., 2015), on the fuel (Christian et al., 2003; Lee et al., 2018) and the stage of the fire (e.g., open fire, smoldering fire (Lee et al., 2010; Popovicheva et al., 2019)). These features will influence for example the ratio of CO/CO2, amount of black carbon emissions as well as the vertical distribution of BC. Enviro-HIRLAM simulations were conducted considering five main aerosol compounds: BC, organic carbon, sulfate, sea salt and mineral dust. The highest values of BC-to-total aerosol content ratio observed in the lower 200-m layer (up to 990 hPa) over fire areas reaching 20% (see Fig.5a). BC normally accounts for 10% at the distance from wildfires by the prevailing wind (Fig. 5d-5i). On the example of Odesa (the most distant city among presented), it is well seen the influence of local urban BC emissions, where BC ratio increased to 15-24% near the surface (see Fig. 5j-5l). Vertical distribution of BC ratio was rather uniform from the surface to the altitude of 2 km ( $\approx$ 800 hPa).

---

## Author Comment (AC4)

**Answer to Reviewer 2**

Dear Anonymous Reviewer,

We are thankful for your constructive criticism and comments to the manuscript "Estimation of elevated black carbon episode over Ukraine using Enviro-HIRLAM" submitted to the ACP journal. Please, see below our replies and changes/ modifications to the original manuscript.

*R2. The authors estimated the elevated black carbon (BC) levels in temporal and three-dimensional spatial scales after the forest fire events occurred in summer 2010 at the center of the European territory of Russia. Throughout the manuscript, the writing, however, appears somewhat rambling, especially for the 'Results and discussion" section. The authors should give more discussion and interpretation for their results.*

A: The Section "Results and Discussion" was revised and re-written accordingly. We expand our analyses with results by adding the analyses of aerosol component ratio, direct aerosol effects on downwelling surface short / long-wave radiation fluxes and air temperature. There was added Aitken mode for previously presented results.

The section 3 "Results and Discussion" was updated with including more details.
In particular, new sub-section "3.3 Ratio of aerosol compounds and observed direct aerosol effects during the wildfire episode" was included with Figures 5-6-7.
"

Aerosol impacts on the atmospheric composition during wildfire events depend on the intensity of the fire (Vadrevu et al., 2015), on the fuel (Christian et al., 2003; Lee et al., 2018) and the stage of the fire (e.g., open fire, smoldering fire (Lee et al., 2010; Popovicheva et al., 2019)). These features will influence for example the ratio of $CO/CO_2$, amount of black carbon emissions as well as the vertical distribution of BC. Enviro-HIRLAM simulations were conducted considering five main aerosol compounds: BC, organic carbon, sulfate, sea salt and mineral dust. The highest values of BC-to-total aerosol content ratio observed in the lower 200-m layer (up to 990 hPa) over fire areas reaching 20% (see Fig.5a). BC normally accounts for 10% at the distance from wildfires by the prevailing wind (Fig. 5d-5i). On the example of Odesa (the most distant city among presented), it is well seen the influence of local urban BC emissions, where BC ratio increased to 15-24% near the surface (see Fig. 5j-5l). Vertical distribution of BC ratio was rather uniform from the surface to the altitude of 2 km ($\approx$800 hPa).

[Figure]

**Figure 5: Vertical distribution of ratio of aerosol compounds in the days of air movement towards Ukraine at 12UTC on 7th August (a,d,g,j), 14th August (b,e,h,k) and 15th August (c,f,I,l).**

Organic carbon very often is the main aerosol component in the lower 2-5-km layer (≈800-550 hPa), especially near the fires, but also at the surface in general (e.g., Jimenez et al. 2009). Based on our results, sulfates prevail throughout the lower and middle troposphere at the distances, and over 3 km (≈700 hPa) level near the wildfires. Hence, aerosol effects over the region were mainly driven by organic carbon and sulfates aerosols. The largest BC contribution during anticyclonic weather conditions seem to be in the boundary layer. In our analysis, mineral dust ratio rarely exceeded 10%, whereas sea salt was detected in the atmosphere only near the sea (see Fig. 5j, 5l).

Based our modeling results, continuous wildfires and aerosol emissions over the region as a consequence influenced the radiative transfer via direct aerosol effects. Enviro-HIRLAM DAE run showed both increase and decrease of downwelling surface long-wave and short-wave radiation. The most intense changes caused by direct aerosol effects were observed during the midday reaching over $\pm 20$ W/m$^2$ for long-wave and over $\pm 100$ W/m$^2$ for short-wave radiation (see example on Fig. 6). The observed direct effects on the downwelling surface radiation co-aligned with the total aerosol impact. Our results indicate that when the BC concentration and the coarse mode exceeded 30 ppb and BC-to-total aerosol mass ratio was over 10% in the lower troposphere, the downwelling surface long-wave radiation tended to increase while the changes of short-wave radiation varied within -200…25 W/m$^2$.

[Figure]

**Figure 6: Differences between DAE and REF for downwelling surface long-wave and short-wave radiation at 12UTC on 14th August 2010.**

In our simulations, direct aerosol effects had an impact on temperature as well. The difference between DAE and REF runs showed that 2-m air temperature during the midday varied from -3°C to 3°C (see Fig. 7). The 2-m air temperature decreased due to the direct aerosol effects prevailed during the wildfire episode. However, the 2-m air temperature was higher with direct aerosol effects in areas where DAE downwelling surface long-wave radiation was higher 12 W/m² and DAE-REF difference for surface short-wave radiation was not lover than -150 W/m². Moreover, the 2-m air temperature was 1-4°C higher at the distance from the wildfires over the territories where the BC coarse mode exceeded mixing ratios of 20-30 ppb.

[Figure]

**Figure 7: Differences between DAE and REF for 2-m Air Temperature at 12UTC on 14th August 2010.**
"

Moreover, the sections 3.1 and 3.2 were also improved by addition more detailed analysis for the Aitken mode (including updated Figures 2-3-4), and in particular, by adding the following text:

For sub-section 3.1:

"At the beginning of August 2010 before the wildfire episode, BC content in Ukraine was low and mostly represented by the accumulation mode with the values of 1-8 ppb. The highest BC concentrations reached 16 ppb (accumulation mode) and was observed at the near-surface level near small local wildfires in the central part of Ukraine. BC Aitken and coarse modes did not exceed 0.5 ppb."

&

"In general, the wildfire emissions have large accumulative effect in the near-surface layer. Total accumulated amount of BC for the period 3–18 August 2010 reached 13500 ppb (for accumulation mode) and 2200 ppb (coarse mode) in the lower tropospheric layer near the burning areas/ cells (Fig. 3). Total accumulated amount of BC Aitken mode caused by wildfires was about 15-30 ppb, whereas the maximum accumulated effects was observed in cities and caused by local anthropogenic emissions (see Aitken mode on Fig. 3). A large amount of combustion products was transported through the atmosphere to the south-west and deposited over territories of the Eastern Ukraine, the Azov and Black Seas. The integral values of BC on these territories exceeded 800 and 150 ppb for accumulation and coarse modes, respectively. This is well seen from Fig. 3 where the regions were affected by intensive deposition processes. Due to smaller sizes of the particles, the accumulation mode has larger spatial coverage and more smooth distribution than the coarse mode."

&

"For the studied period, in general, ground-based dust measurements also showed elevated levels connected with forest fires. Almost all geographical regions except northern part of Ukraine have clear maxima of ground-based dust concentrations during August 2010. The highest exceeding over urban background values in August were observed on the eastern territories. Here, the integral value of the near-surface BC concentrations for the coarse mode was higher than 250 ppb. For the cities on the east, the dust concentrations were 27–47% higher than average dust content in 2010. Moreover, these were also 23–72% higher than multi-year average concentrations for month of August. Overall dust content on the east was 0.02–0.07 $\mu g/m^3$ higher than usually in August.

Large difference in dust content between August and other months was observed on the seashore of the Azov Sea and in the central part of Ukraine. The concentrations were higher on 0.05–0.25 $\mu g/m^3$ than usually in the same month. Majority of cities in the central part of Ukraine showed 17–73% higher dust concentrations than average in 2010 and 8–45% higher than usually in August. In the western parts of Ukraine, the integral values were lower than 100 and 500 ppb for coarse and accumulation modes, respectively (Fig. 3)."

for sub-section 3.2:

"In contrast to the BC coarse and accumulation modes, Aitken mode was observed throughout the entire troposphere up to 200 hPa level (Fig. 4) height, however the wildfire emissions did not prevail over the anthropogenic emissions. The BC Aitken mode rarely exceeded 0.2 ppb at 950 hPa level. Therefore, concentrations more than 1 ppb were observed only near the surface and had clear maximum over urban areas (see Fig. 3). This is the reason why the BC Aitken mode near the surface was higher in more distant Odesa than in Kalac region"

& sub-section 3.4 "Discussion" was added:

"BC dispersion and vertical distribution during stationary anticyclonic conditions had their own specific features. The most typical are uniform vertical profile up to 700 hPa (≈3 km) and BC ratio of 8-20% among all aerosol compounds. Very often there is no clear maximum within lowest 3-km layer especially for coarse and accumulation mode. Urban areas with additional anthropogenic emissions are the exception where maximum formed near the surface. This distribution differs from other synoptic conditions or its averaging, which results in maximum BC concentration outside the emission areas at an altitude of 1.5 km (Kostrykin et al., 2021).

The decrease of radiation forcing (Chubarova et al., 2012) and reduction of shortwave radiation (Pere et al, 2014) were observed after wildfire aerosol emissions during summer 2010. Previously it was shown that shortwave radiation is smaller up to 70-84 $W/m^2$ in diurnal averages (Pere et al., 2014). Our results confirmed the decrease of downwelling surface short-wave radiation at the background as the result of direct aerosol effects up to -250 $W/m^2$ during the midday hours. However, numerous spots were observed where downwelling surface short-wave and long-wave radiation increased. 2-m air temperature is higher considering direct aerosol effects in areas where long-wave radiation increased up to 20 $W/m^2$ and both BC coarse and accumulation mode exceeded 30 ppb. These increasing might correspond to BC impact at certain locations, as BC influence is known by its localizing effects with no clear directs link between the pattern of forcing and pattern of temperature change (Modak and Bala, 2019). Overall, direct radiative forcing and air temperature increase was estimated for BC regardless emission source (Ma et al., 2018; Zhuang et al., 2019; Kostrykin et al., 2021); however cooling effects also was detected (Ma et al., 2018). In combination with other aerosol types the effects on radiative forcing also is different (Kirkevag et. al, 1999). Hence, not all areas with air temperature increasing or decreasing could be directly connected with the prevailing influence of certain aerosol compound.

The absence of BC measurements at ground-based monitoring network in Ukraine represents a challenge for results validation. Nevertheless, it gives overall picture on BC distribution during unfavorable weather conditions and makes possible to provide assessments in Ukraine on how BC influenced human health and local ecosystems. Moreover, the fact that wildfire emissions could cause local air temperature increasing amid elevated pollution levels is crucial for understanding the consequences for vulnerable people. This is especially relevant during the heat wave events likewise it was in August 2010 discussed in this study.
"

*R2. The authors need to explain exactly what the novel insights of their work are, and how their finding are relevant to atmospheric chemistry and radiative transfer.*

A: The Abstract, sections 3 "Results and Discussion" and "Conclusions" were revised and improved by including key findings of this study, and in particular, this study in Abstract:

"BC contribution was found to be 10-20% among total aerosol compounds near the wildfires in the lower 2-km layer. At the distance, BC exceeded 10% only over the urban areas. In the areas with high BC content represented by both accumulation and coarse mode, downwelling surface long-wave radiation increased up to 20 $W/m^2$ and 2-m air temperature increased on 1-4°C during the midday hours."

In addition, "This study is the first for Ukraine which describe elevated spatio-temporal BC content in the three-dimensional scale, distribution of its different particle sizes and BC ratio among other aerosol compounds in the atmosphere with the emphasize on extremely hot period.".

Moreover, for the first time, the seamless online integrated meteorology - atmospheric composition modelling approach (Enviro-HIRLAM model), compared to classical off-line approach (separate meteorology and atmospheric chemical transport modelling), was applied to geographical domain of Ukraine.

*R2. Especially compared with the earlier works that investigated the forest fire events occurred in summer 2010, the authors could state what questions had been left open, and what the present paper now addresses.*

A: The section "Introduction" was updated by including the following relevant publications analysis: "In contrast to other the aerosol compounds, BC typically cause radiative a positive forcing (Bond et al., 2013; Stjern et al., 2017) which intensity depends on the particle size (Matsui et al., 2018). Consequently, the heating effect is generally observed from wildfire emissions (Kostrykin et al., 2021), and also from anthropogenic sources (Zhuang et al., 2019). Sometimes a cooling effect was also detected (Ma et al., 2018). All these effects, however, are localized (Modak and Bala, 2019)."

&

"Described wildfire events frequently occur in Ukraine or in territories of neighboring countries that develop into elevated pollution events. Unfortunately, the existing official Ukrainian air quality monitoring network (UA-AQMN) does not provide any measurements of BC. Hence, the modelling remains the only possible way to estimate the spatio-temporal variability of BC in Ukraine, and to explore the consequences of elevated pollution episodes and assess the short-term impacts in the region. The lack of observation capacity on BC on a national level caused a gap in knowledge on how BC is distributed; what are the impacts on local ecosystems; and which mitigation measures are needed to improve the situation. This study is the first in Ukraine which describes the elevated spatio-temporal BC content on the three-dimensional scale, explores the distribution of BC in particle sizes and compares the BC ratio between other aerosol components in the atmosphere with the emphasize on extremely hot weather episodes. Furthermore, we aimed to estimate the impact of wildfire emissions on the surface by considering direct aerosol effects, as the response of radiative and temperature regimes varying in different regions and depending on the ratio of aerosol components."

&

additionally, analyzed in this study "Direct aerosol effects on downwelling surface short-wave/ long-wave radiation and 2-m air temperature also are discussed."

&

in sub-section 3.4 "Discussion"

"
The BC dispersion and vertical distribution during stationary anticyclonic conditions had their own specific features. The most typical cases were a uniform vertical profile up to 700 hPa (≈3 km) with a BC ratio of 8-20% among all the aerosol compounds. Very often there was no clear maximum

within the lowest 3-km layer especially for the coarse and accumulation modes. The urban areas with additional anthropogenic emissions were an exception where a maximum formed near the surface. This distribution differed from the other synoptic conditions, which resulted in a maximum BC concentration outside the emission areas at an altitude of 1.5 km (Kostrykin et al., 2021).

The decrease of radiation forcing (Chubarova et al., 2012) and a reduction of shortwave radiation (Pere et al, 2014) were observed after wildfire aerosol emissions during summer 2010. Previously it was shown that the shortwave radiation is smaller on 70-84 W/m$^2$ in the diurnal averages (Pere et al., 2014). Our results confirmed the decrease in the downwelling surface short-wave radiation at the background as the result of the direct aerosol effects were up to -250 W/m$^2$ during the midday hours. However, numerous spots were observed, where the downwelling surface short-wave and long-wave radiation increased. The 2-m air temperature was higher in spite of considering the direct aerosol effects of the areas where the long-wave radiation increased up to 20 W/m$^2$ and both BC coarse and accumulation mode exceeded values of 30 ppb. These hot-spots might coaligned with the BC impacts at certain locations, as the BC influence is known by its localizing effects with no clear directs link between the pattern of forcing and pattern of temperature change (Modak and Bala, 2019). Overall, the direct radiative forcing and the consequent air temperature increase was estimated for BC regardless of the emission source (Ma et al., 2018; Zhuang et al., 2019; Kostrykin et al., 2021); however cooling effects also was detected (Ma et al., 2018). In combination with other aerosol types the effects on the radiative forcing also is variable (Kirkevag et. al, 1999). Hence, not all areas with air temperature increasing or decreasing are not directly connected with the prevailing influence of certain aerosol compound but a net effect including physical, chemical and meteorological effects are being experienced.

In the absence of BC measurements, ground-based monitoring network in Ukraine represents a challenge for the results validation. Nevertheless, it gives an overall picture of the BC distribution during unfavorable weather conditions and makes it possible for a provision of assessments in Ukraine on the quantify of BC influenced human health and local ecosystems in Ukraine as a whole. Moreover, the fact that the wildfire emissions can cause local the air temperature to increase amid elevated pollution levels is crucial for the understanding the consequences for vulnerable people. This is especially relevant during the heat wave events such as in August 2010, which is discussed in this study."

& in Section "Conclusions":

"Overall, the study needs further development in two directions. First direction requires elaboration of ground-based BC measurements in Ukraine for modelling data validation and assessment the negative BC impact on human health and local ecosystems in Ukraine with future transition to mitigation measures and reduction strategy. Second direction should expand the analysis of direct and indirect aerosol effects using online-integrated modelling with separating the influences of different aerosol compounds."

***General Comments***:

*R2. The authors claimed that the work focused on the horizontal and vertical variability of BC concentrations over Ukraine during a wildfire episode in August 2010. However, the author just presented the BC concentrations after the forest fire evens, not comparing with BC levels before the wildfire evens; how to demonstrate that the BC levels were elevated?*

A: Information about BC concentrations before wildfire episode was added to the Section 3.1 "Synoptic weather situation and the dispersion of wildfire emissions during August 2010 in Ukraine". The following text was added:

"At the beginning of August 2010 before the wildfire episode, the BC content over Ukraine was low and mostly it was represented by the accumulation mode with the values of 1-8 ppb. The highest BC concentrations reached 16 ppb (accumulation mode) and was observed at the near-surface level near small local wildfires in the central part of Ukraine. The BC concentrations in the Aitken and coarse modes did not exceed 0.5 ppb."

*R2. Additionally, the authors should valid their simulations of BC concentrations. Could they provide some BC measurements rather than dust observations to verify their simulations and discussed the uncertainties of the simulations.*

A: Unfortunately, this is a big and serious problem for Ukraine. There are no monitoring sites which can provide black carbon (BC) measurements (and also many others chemical species). Hence, the modelling remains the only available tool for the Ukrainian territory, which allowed to estimate the spatio-temporal distribution of such species like BC, organic carbon, etc. As a result, there is a gap in scientific knowledge about BC in Ukraine, and there are no assessments how it is distributed; how it influences local ecosystems, human health; and, consequently, no relevant mitigation measures were taken.

Moreover, the existing official Ukrainian air quality monitoring network (UA-AQMN) was developed and established in the former USSR, and it was not significantly improved or upgraded during the last decades. The monitoring sites (more than 120 in total) of the network do not measure PM; and dust – is the only pollutant (measured by official air quality stations) which could describe aerosol content. We add details on it in the "Methods"-section, including what "dust" is and meaning according to existing local regulations in Ukraine. Of course, the process for EU Directives implementation has been already started in Ukraine, but it is still in slow progress and not finished yet.

As soon as BC measurements will be launched as part of UA-AQMN, it would be possible to study it accurately and precisely. But nowadays only the modelling approach could be used. Nevertheless, first assessments are necessary for interests and needs of Ukrainian national economy following UN Sustainable Development Goals, and in particular, SGDs 3, 9, 13.

We described and explained this in the revised version of the manuscript.

Accordingly, the following changes/additions were incorporated in the revised manuscript: in Section "Introduction":

"Described wildfire events frequently occur in Ukraine or in territories of neighboring countries that develop into elevated pollution events. Unfortunately, the existing official Ukrainian air quality monitoring network (UA-AQMN) does not provide any measurements of BC. Hence, the modelling remains the only possible way to estimate the spatio-temporal variability of BC in Ukraine, and to explore the consequences of elevated pollution episodes and assess the short-term impacts in the region. The lack of observation capacity on BC on a national level caused a gap in knowledge on how BC is distributed; what are the impacts on local ecosystems; and which mitigation measures are needed to improve the situation. This study is the first in Ukraine which describes the elevated spatio-temporal BC content on the three-dimensional scale, explores the distribution of BC in particle sizes and compares the BC ratio between other aerosol components in the atmosphere with the emphasize on extremely hot weather episodes. Furthermore, we aimed to estimate the impact of wildfire emissions on the surface by considering direct aerosol effects, as the response of radiative and temperature regimes varying in different regions and depending on the ratio of aerosol components."

& in sub-section 2.2 "Additional data for analysis"

"Unfortunately, UA-AQMN does not provide measurements of BC, moreover PM also is not measured. Dust – is the only available pollutant which could be compared to the modeled aerosol species. The dust is measured at more than120 monitoring sites, and measurements contain all coarse aerosol particles regardless of their origin (Nadtochii et al., 2019). Dust concentration on these sites is measured using the method of weighing the total suspended particles (RD, 1991)."

*R2. The authors claimed that the features of BC distributions that they presented in this work not only resulted from biomass burning, but also were affected by the local sources of fires. Could they estimate how much the influence?*

A: We accurately evaluated information about sources of local fires for studied period in Ukraine and hence, re-evaluated results and excluded Fig. 7.

**Specific comments:**

*R2. Overall, the paper is rather poorly written with many grammar/ formatting mistakes.*
A: The manuscript was revised following remarks/comments, portions of the text were re-written, grammar were re-checked and we hope that the manuscript text improved.

*R2. Some simple things, such as the superscripts and subscripts of "PM10" "PM2.5" and "µg/m3" (several times),*
A: Superscripts and subscripts were checked and corrected accordingly.

*R2. inconsistent format of the figure legends (font size and line space (caring Figure 1), font bold (caring Figure 2)).*
A: Figures' legends were checked and corrected accordingly. Moreover, all illustrations/ figures were re-drawn and higher resolution and clarity.

*R2. These mistakes indicate that proper care was not taken to proofread the paper prior to submission.*
A: Mistakes were corrected following remarks/comments, some text/paragraphs of the manuscript were re-written, grammar re-checked, additional proof-reading was also performed. By doing this, we hope that the manuscript text became of better quality.

*R2. Figure 2 is from the Climate Forecast System (CFS) Reanalysis (source: www.wetterzentrale.de). It can be shown in the Supporting Information.*
A: Figure 2 moved to Appendix A.

*R2. The "3.1 Synoptic weather situation during summer 2010 in Ukraine" just has one paragraph, which can be combined with the "3.2 Dispersion of wildfire emissions".*
A: Section 3.1 was combined with 3.2 and 3.4. We entitled the Section "3.1 Synoptic weather situation and the dispersion of wildfire emissions during August 2010 in Ukraine". Information about BC content before wildfire event was also added to this Section.

*R2. Page 7 /Line 160-164: The authors discussed the horizontal distribution of BC concentrations not only in the accumulation mode but also in the coarse mode. However, the authors only showed the BC distribution in the accumulation mode in Figure 3. Why not present the simulations of coarse mode.*

A: The BC distribution for Aitken and coarse modes was added in Figure 2 (Fig. 3 in previous version of the manuscript).

And additionally, the following new text included "The highest values for BC Aitken mode did not exceed 6 ppb near the fires and the simulated values were lower than 2 ppb at distances from the burning areas. Moreover, the highest values of Aitken mode were intermittently observed over urban areas reaching 10 ppb.".

*R2. Page 9 /Line 195: The authors highlighted that the elevated BC concentrations were detected at 590 hPa (for the coarse mode) and at 550 hPa (for the accumulation mode). However, Figure 5 shows that the BC concentrations were very low with values near 0 in the <650 hPa layers. Could the authors clear how they identify the elevated BC concentrations at 590 hPa (for the coarse mode) and at 550 hPa (for the accumulation mode).*

A: We improved our Figure 4 (Fig. 5 in previous version of the manuscript), Aitken mode also included. Now, the heights become visible, where BC was detected during the wildfire event. Additionally, the following new text was added:

"In contrast to BC coarse and accumulation mode, Aitken mode was observed throughout the entire troposphere up to 200 hPa level (Fig. 4), however wildfires did not prevail over anthropogenic emissions. BC Aitken mode rarely exceeded 0.2 ppb over 950 hPa level. Concentrations more than 1 ppb were observed only near the surface and had clear maximum over urban areas (see Fig. 3). This is the reason why BC Aitken mode near the surface is higher in more distant Odesa than in Kalac."

*R2. Page 10 /Line 219-225: This paragraph introduced how to process the ground-based dust measurements. It should be in the "Data and methods" rather than the "Results and discussion".*

A: We re-wrote original text and moved to section 2 "Data and methods". In particular, the revised text is the following:

"Unfortunately, UA-AQMN does not provide measurements of BC, moreover PM also is not measured. Dust – is the only available pollutant which could be compared to aerosol species. Dust is measured at more than 120 monitoring sites, and measurements contain all coarse aerosol particles regardless of their origin (Nadtochii et al., 2019). Dust on these sites is measured using the method of weighing the coarse aerosol particles (RD, 1991) with the size not less than $PM_{10}$ fraction."

*R2. Page 11 /Line 239-241: The two sentences are confusing and needs to be reworded.*
A: Confusing sentences were removed/ rewritten. See our answer on your question in your next question and our answer to that.

*R2. Page 12 /Line 255-259: The authors discussed the "tails" of the BC distribution in the coarse mode, attributing to atmospheric transport and the impact of local fires. Why the BC distribution in*

*accumulation mode did not present "trail" feature. How the effects of the atmospheric transport and local fires on the BC distribution in the accumulation mode?*

A: We change the color scale of the BC accumulation mode in Fig. 3 (fig 5 in the previous version of the manuscript). Also, we reconsider the results and removed the explanation about the possible impact of local fires. We combined this Section with Section 3.1. In particular, the revised figure is now as follows:

[Figure]

**Figure 3: The integral value of the near-surface BC concentrations for the Aitken, accumulation and coarse modes for the period 3-18 August 2010.**

& the revised text is the following:

"In general, the wildfire emissions have a large accumulative effect in the near-surface layer. Total accumulated amount of BC for the period 3–18 August 2010 reached 13500 ppb (for the accumulation mode) and 2200 ppb (for the coarse mode) in the lower tropospheric layer near the burning areas (Fig. 3). The total accumulated amount of BC Aitken mode caused by wildfires was about 15-30 ppb, whereas the maximum accumulated impacts were observed in the cities (see e.g., Aitken mode on Fig. 3). A large amount of combustion products was transported through the atmosphere to the south-west and deposited over the territories of the Eastern Ukraine, the Azov and Black Seas. The integral values of BC on these territories exceeded values of 800 and 150 ppb for the accumulation and for the coarse modes, respectively. This is well seen in Fig. 3 where the regions were affected by intensive deposition processes. Due to the smaller sizes of the particles, the accumulation mode had a larger spatial extent and more smooth distribution than the coarse mode.

For the studied period, in general, the ground-based dust measurements also showed elevated levels connected with the forest fires. Almost all the geographical regions except northern part of Ukraine experienced a clear maxima of ground-based dust concentrations during August 2010. The highest exceeding over the urban background values in August were observed on the eastern territories. In this case, the integral value of the near-surface BC concentrations for the coarse mode was higher than 250 ppb. For the cities on the east, the dust concentrations were 27–47% higher than the average dust content in 2010. Moreover, these values were also 23–72% higher than multi-year average concentrations for the month of August. Overall, the dust content on the east Ukraine was 0.02–0.07 μg/m³, which was higher than usually occurred in August.

Large difference in the dust content between August and other months was observed on the seashore of the Azov Sea and in the central part of Ukraine. The concentrations were higher by 0.05–0.25 μg/m³ than usually in the same month. A majority of cities in the central part of Ukraine showed 17–73% higher dust concentrations than average in 2010 and concurrently 8–45% higher than usually in August. In the western parts of Ukraine, the integral values were lower than 100 and 500 ppb for coarse and accumulation modes, respectively (Fig. 3)."

*R2. Conclusions: The authors need to clear the novel insights of their work and the significance of their finding.*

A: The section 4 "Conclusion" was re-written to underline more details on findings of this study, by including the following text:

"The highest BC-to-total aerosol content ratio was observed in the lower 200-m layer (up to 990 hPa) reaching 10-24% near the fires and over urban areas. The dominant aerosol compounds were organic carbon and sulfates. Hence, direct aerosol effects on radiative and temperature regimes in the region mostly were typical for these compounds: decreasing in downwelling surface short-wave radiation and decreasing of 2-m air temperature. In areas with high BC content representing by accumulation and coarse mode, downwelling long-wave radiation with direct aerosol effects was higher up to 20-25 W/m$^2$ during the midday hours. 2-m air temperature was 1-4°C higher in these regions with BC coarse mode exceeded 20-30 ppb.".

Additional text was also included "For the first time, spatio-temporal BC content, distribution of its different particle sizes, and BC ratio among other aerosol compounds in the atmosphere with the emphasize on extremely hot period were analyzed."

"Moreover, for the first time, the seamless online integrated meteorology - atmospheric composition modelling approach (with Enviro-HIRLAM model), compared to classical off-line approach (separate meteorology and atmospheric chemical transport modelling), was applied to geographical domain of Ukraine."

Plus, extra text added was added to the paper as follows:

"The obtained results of the study are important and significant for improvement of quality of numerical weather prediction and for depicting the impacts of the extreme events with atmospheric chemical transport modelling. The findings are relevant for assessment studies of atmospheric pollutants impact on population and ecosystems health, for climate adaptation and socio-economical related studies, for optimization and establishing of air quality monitoring stations, decision- and policy making process, etc."

---

## Author Response (AR2)

**Reviewer 1**

Dear Anonymous Reviewer,

We are thankful for your constructive criticism and comments to the manuscript "Estimation of elevated black carbon episode over Ukraine using Enviro-HIRLAM" submitted to the ACP journal. Please, see below our replies and changes/ modifications to the manuscript.

***R1. It is appreciated that the authors have made great efforts to revise the previous manuscript. There is some more information added and the manuscript is improved. However, there are still large room for further improvement for its consideration in publishing in ACP. I only list a few points which may help, but the overall writing still needs considerable improvement.***
**A**. Following your comments/ suggestions, the manuscript was further improved. Please, see detailed replies below.

***R1. Why the unit of BC is ppb? Why not give mass concentration, are you modelling the number concentration of BC? I only see the unit of aerosol concentration is mass per unit volume (or mass) of air.***
**A.** The Enviro-HIRLAM output of aerosol compounds is given in ppb. For clarification, as the obtained modelling results are the only available data of BC in Ukraine, we used the ppb units for BC, and to avoid confusion, we converted ground-level measurements of dust to ppb as well in order to unify our comparison.

***R1. The map in Fig. 1 could be colored with terrain height for a better demonstration.***
**A**. Fig. 1 was updated by including terrain relevant data to underline presence of the mountainous areas.

***R1. The reason in selecting the three sites should be more explicitly given. The three targeting places should be also marked in the maps in Fig. 2-4. The reason in selecting these three sites needs to be explained by careful analysis of synoptic meteorology.***
**A**. We clarified in Section 2.2 as (Lines 158-165):
"The air temperature vertical profiles were analyzed for 2–18 August 2010 at the following sounding stations: Kyiv (station code 33345), Kharkiv (34300), Odesa (33837), Rostov-na-Donu (34731), Kalac (34247) and Voronezh (34122). Unfortunately, there is a limited number of the sounding stations in the region of interest. Hence, Voronezh and Kalac were selected because these are situated within a radius of less than 400 km away from the forest fires area (being the nearest sounding stations to wildfires). Kharkiv and Rostov-na-Donu were selected for estimation of temperature inversion impact at some distance (within a radius of up to 600 km) away from the fires. Kyiv and Odesa were chosen as these are relatively distant (within a radius of up to 1000 km) to wildfires emission sources. The air temperature vertical profiles were considered for up to 3.5 km above the ground surface at 00 UTC."

Overall, we analyzed BC distribution in all grid cells over the region of interest (i.e. not only for these selected sites). The analysis and explanation in the section "Results" refer to all region of interest. In the same way, for Fig. 2 and Fig. 3 – analysis was made for the entire geographical territory, not for the specific locations. These sites are the sounding stations, and we used these locations in "Results" while describing vertical profiles in Fig. 5 considering in-situ aerological measurements. Air temperature profiles were analyzed for these 5 stations, but on plots we showed three of them as an example (one station depending on the distance).

The description of synoptic situation was amended as follows:

"…a blocking anticyclone caused severe hot weather, lack of precipitation, which lead to the occurrence of wildfires. The period lasted from the end of June to the second half of August 2010 over Eastern Europe and the south-western regions of Russia (see example Fig. A1). Hot air masses from Central Asia penetrated the territories on the north-west, and the anticyclone was detected throughout the whole troposphere before the highest pollution levels were transported and distributed out of burning areas. Continuous extreme weather and clear sky conditions together with the highest insolation in the middle latitudes caused the dominance of high temperature and low humidity regimes. These led to the most favourable conditions for drought formation that played a crucial role in initiating the wildfires and contributed to their rapid spread." (Lines 169-175)

"The observed anticyclonic conditions influenced the formation and development of spatio-temporal patterns for BC atmospheric transport and dispersion. The time-series for each grid point consisted of two maxima. These were related to the observed atmospheric circulation patterns. A typical clockwise air movement for anticyclones in the Northern Hemisphere caused an intensive atmospheric transport towards Ukraine during two periods: 7-8 and 13-16 August 2010." (Lines 183-186)

"The dominated hot and dry weather conditions interrupted in the second half of August 2010. It occurred when the blocking anticyclone weakened, and a cyclone arrived from the western sector." (Lines 204-205)

*R1. The tile should be revised, what "BC episode" mean?*
**A**. From definition, the using of word "episode" is attributed to an event or a group of events occurring as part of a sequence. The wording as "air pollution episode" is frequently used as well. BC is one of air pollutants, and we are using the wording "BC episode", when we discuss the episodic elevated BC concentration events in Ukraine.
To clarify, we suggest to edit the current title – "Estimation of elevated black carbon episode over Ukraine using Enviro-HIRLAM" to "Enviro-HIRLAM model estimates of elevated black carbon pollution over Ukraine resulted from forest fires".

*R1. It is not clear throughout the text whether the presented BC concentration is columnar concentration (layer-integrated) or surface concentration. In Fig. 3, what is the integral near-surface?*
**A**. BC concentration was modelled at 40 model levels. These details are given in Section 2.1 (Lines 116-117):
"The vertical structure included 40 model levels (up to 10 hPa) with a more detailed resolution in the boundary layer with 22 model levels from the surface up to 500 hPa. This provided a great opportunity to study the BC vertical atmospheric transport."
We analyzed BC concentration both for near-surface level and at the heights/ altitudes, but we did not analyze integrated columnar content.
For example, caption to Fig. 2 contains: "…with the highest BC content near the surface", to Fig. 3 "The integral value of the near-surface BC concentrations…". We identified pressure levels in the text when we analyzed BC content at the heights.
In the revised version, we have clarified throughout the text where it was not accurately mentioned.
The explanation of integral near-surface content is given in the Section 2.1 (Lines 139-143). It was clarified as the following:
"The spatial analysis of model output was carried out considering all grid cells without spatial averaging and interpolation. It enabled the detection of concentration changes within each grid caused by anticyclonic air movements. Evaluation of accumulated BC impact was performed by time

integration of near-surface BC concentration at the lowest (near-surface) model level for the studied period. Therefore, the summed values represent the total amount (for the Aitken, accumulation, and coarse modes in ppb) transported by air movements through grid cells near the surface."

There are also more details in the Section 3.1 (Lines 207-215):
"In general, wildfire emissions had a large cumulative effect in the near-surface layer concentrations. The total accumulated amount of BC for the period 3–18 August 2010 reached 13500 ppb for the accumulation mode and 2200 ppb for the coarse mode, respectively, in the lower tropospheric layer near the burning areas (Fig. 3). The total accumulated amount of BC Aitken mode caused by wildfires was about 15-30 ppb, whereas the maximum accumulated impacts were observed in the cities (see e.g., Aitken mode in Fig. 3). A large amount of combustion products was transported through the atmosphere to the south-west and deposited over territories of the Eastern Ukraine, the Azov and Black Seas. The integral values of BC for these territories exceeded values of 800 and 150 ppb for the accumulation and for the coarse modes, respectively. This is seen in Fig. 3, where the regions were affected by intensive deposition processes. Due to the smaller sizes of the particles, the accumulation mode had a larger spatial extent and a smoother distribution than the coarse mode."

***R1. What is the difference between Fig. 2 and Fig. 3. What do you mean by "during the episode when highest surface concentration"?***
**A.** We corrected caption to Fig. 2 as: "Figure 2: The spatial distribution of the BC for the Aitken (a,d,g), accumulation (b,e,h) and coarse (c,f,i) modes in the days of air movement towards Ukraine with the highest BC content near the surface."
Fig. 2 shows BC content at specific time and day with the highest BC content near the surface (when air masses moved towards Ukraine). Fig. 3 shows BC content near the surface integrated with time (it was done to estimate accumulative effect in the near-surface layer which has the largest impact for human health).

***R1. It is unnecessary to show the sub-figures in Fig. 4, you may just use a log x-axis. The x-axis needs to be consistent among figures to be comparable.***
**A.** The Fig. 4 was corrected.

***R1. There are a lot of information which has not been conveyed clearly in Fig. 4. How the accumulation mode BC had changed so dramatically? This needs to be presented in a map along with synoptic analysis and fire points to show 1) how BC has been transported 2) how the particle size had evolved. The key question needs to be answered: how the size of BC and mass loadings has been changed. I still don't get the reason why BC could be so large for biomass burning, are those mixed with dust?***
**A**. After we added new Fig. 4 and revised the text, the Sections 3.1 and 3.2 together with Figures 2-5 respond these questions:

[revised manuscript text omitted]